# A non-canonical ARMS-GABARAP interaction modulates dendritic spine formation and synaptic development

Wenli Jiang[1,2], Jin Ye [ID][1,5 ✉], Jiasheng Chen[1,2], Xinyu Wang [ID][1,2], Yahong Li[1,2], Jianchao Li [ID][3], Yide Mei [ID][2], Yanlu Lyu [ID][4 ✉], Wei Hu [ID][1 ✉] & Chao Wang [ID][1,2 ✉]

## Abstract

**ARMS (ankyrin repeat-rich membrane spanning) is a scaffold protein essential for neurotrophic signaling, synaptic development, and cytoskeletal remodeling. Despite its central role in neuronal function, how ARMS is regulated at the molecular level remains poorly understood. Here, we identify GABARAP, an Atg8-family autophagy adaptor, as a novel ARMS-binding protein that directly interacts with its N-terminal ankyrin repeats. We present the crystal structure of the ARMS-GABARAP complex, revealing an atypical interaction mode distinct from canonical LIR-dependent Atg8 interactions. Remarkably, ARMS specifically binds to the GABARAP subfamily of Atg8 proteins, setting it apart from the LC3 subfamily. Functional analysis demonstrates that GABARAP negatively regulates ARMS-mediated dendritic spine development and maturation in hippocampal neurons. Additionally, disrupting the ARMS-GABARAP complex using ankyrin-derived peptides alters ARMS subcellular localization, increasing its accumulation in the soma of neurons. Collectively, our findings uncover a novel ARMS-GABARAP interaction mechanism, establish the regulatory role of this complex in neuronal protein homeostasis, and suggest potential therapeutic strategies for targeting scaffold protein interactions in neurodevelopmental and neurodegenerative disorders.**

**Keywords** Autophagy; Synaptic Development; Neuronal Protein Homeostasis; Scaffold Proteins; Protein-protein Interaction
**Subject Categories** Autophagy & Cell Death; Translation & Protein Quality

## Introduction

Neuronal development and synaptic plasticity rely on precisely coordinated intracellular signaling and cytoskeletal organization (Conde and Cáceres, 2009; Wu et al, 2023; Nanou and Catterall, 2018). Scaffold proteins play central roles in these processes by acting as molecular hubs that integrate signaling pathways, regulate protein trafficking, and organize macromolecular complexes at specialized neuronal compartments (Cesca et al, 2015; Cunha et al, 2009). One such scaffold protein, ankyrin repeat-rich membrane spanning (ARMS, also known as Kidins220), has emerged as a critical regulator of neuronal survival (Iglesias et al, 2000), synapse formation (Wu et al, 2009), and cytoskeletal remodeling (Jung et al, 2014; Riol-Blanco et al, 2004). Loss of ARMS function severely disrupts neuronal development, with homozygous knockout mice exhibiting embryonic lethality (Cesca et al, 2012), and heterozygous mice displaying profound dendritic abnormalities (Albini et al, 2024; Chen et al, 2012).

ARMS is highly enriched in neurons and serves as a downstream scaffold for neurotrophin signaling pathways (Arévalo et al, 2004), integrating signals from receptor tyrosine kinases (RTKs), including Trk receptors (Chang et al, 2004), and modulating intracellular cascades essential for neuronal polarization, axonal guidance, and dendritic arborization (Kong et al, 2001; Arévalo et al, 2006). At the synaptic level, ARMS has been shown to influence glutamatergic synapse development (Sutachan et al, 2010; Neubrand et al, 2012), promote dendritic spine stability, and facilitate synaptic plasticity by interacting with key signaling molecules (Sniderhan et al, 2008; Cortes et al, 2007). Dysregulation of ARMS has been implicated in neurological disorders such as Alzheimer's disease and synaptic integrity defects (Lopez-Menendez et al, 2013; Puerto et al, 2023), underscoring its importance in maintaining neuronal homeostasis (Puerto et al, 2021; Almacellas-Barbanoj et al, 2022). Despite its critical role in neuronal function, the mechanisms that regulate ARMS stability, subcellular localization, and interaction with intracellular partners remain largely unknown. Given that scaffold proteins often

[1]Department of Neurology, the First Affiliated Hospital of USTC, Ministry of Education Key Laboratory for Membraneless Organelles and Cellular Dynamics, School of Life Sciences, University of Science and Technology of China, 230027 Hefei, China. [2]Hefei National Research Center for Physical Sciences at the Microscale, Biomedical Sciences and Health Laboratory of Anhui Province, Center for Advanced Interdisciplinary Science and Biomedicine of IHM, Division of Life Sciences and Medicine, University of Science and Technology of China, 230027 Hefei, China. [3]Division of Cell, Developmental and Integrative Biology, School of Medicine, South China University of Technology, 510006 Guangzhou, P.R. China. [4]Department of Otorhinolaryngology, Shenzhen People's Hospital (The First Affiliated Hospital, Southern University of Science and Technology; The Second Clinical Medical College, Jinan University), 518055 Shenzhen, China. [5]Present address: School of Life Sciences, East China Normal University, Shanghai, China. ✉E-mail: jinye123@ustc.edu.cn; drlv_ent@outlook.com; andinghu@ustc.edu.cn; cwangust@ustc.edu.cn

function within dynamic macromolecular complexes, it is unclear whether ARMS requires additional regulatory factors to modulate its function in synapse formation and neuronal signaling.

GABARAP (γ-aminobutyric acid receptor-associated protein) is a member of the Atg8 family of autophagy-related proteins, originally identified as a trafficking regulator for GABA$_A$ receptors (Ye et al, 2021; Hui et al, 2019). Beyond receptor trafficking, GABARAP plays essential roles in synaptic organization (Chen et al, 2000), cytoskeletal remodeling (Nelson et al, 2020), and intracellular transport pathways (Johansen and Lamark, 2020). As a key component of the selective autophagy machinery, GABARAP facilitates cargo recognition and autophagosome maturation, linking synaptic function to protein turnover and degradation pathways (Johansen and Lamark, 2020; Schaaf et al, 2016; Birgisdottir et al, 2013). A defining feature of GABARAP-mediated interactions is its recognition of LC3-interacting region (LIR) motifs, which engage the two hydrophobic pockets (HP1 and HP2) on GABARAP, forming a canonical LIR docking site (LDS) interaction (Wesch et al, 2020; Birgisdottir et al, 2019; Bavro et al, 2002). Given that ARMS contains predicted LIR-like sequences within its ankyrin repeat domain, it raises the question of whether ARMS directly interacts with GABARAP and, if so, how this interaction contributes to its neuronal functions. To date, no studies have explored the potential regulatory interplay between ARMS and GABARAP.

In this study, we identify GABARAP as a novel binding partner of ARMS, establishing a direct interaction mediated by the ankyrin repeats (ARs) of ARMS, rather than the classical LIR motif. We determine the high-resolution crystal structure of the ARMS 1-4 ARs-GABARAP complex, revealing a previously unrecognized binding mode in which ARMS finger-loop regions engage GABARAP hydrophobic pockets via a combination of electrostatic and hydrophobic interactions. Notably, this interaction is highly specific to the GABARAP subfamily of Atg8 proteins. Functionally, we demonstrate that ARMS promotes neurite outgrowth and dendritic spine maturation, whereas GABARAP negatively regulates these processes. Furthermore, GABARAP facilitates ARMS degradation through an autophagy-dependent mechanism, selectively preventing ARMS accumulation beyond physiological levels. Disrupting the ARMS-GABARAP interaction using highly specific ankyrin-derived peptides alters ARMS subcellular localization, revealing a previously unknown mechanism by which GABARAP modulates ARMS stability and function. Together, our findings define a novel, non-canonical ARMS-GABARAP interaction and establish its functional significance in neuronal development, synaptic organization, and protein homeostasis. These results provide new insights into the mechanisms by which scaffold proteins coordinate neuronal signaling and suggest broader implications for GABARAP-mediated regulation of synaptic proteins.

# Results

## The ankyrin repeats of ARMS directly interacts with GABARAP

To explore the function of ARMS, we sought to identify its novel interaction partners. Given that most GABARAP subfamily proteins interact with their binding partners via one or more LIR motifs (Johansen and Lamark, 2020), we first predicted whether

ARMS contained a canonical LIR motif and could interact with GABARAP. Bioinformatics analysis revealed two putative LIR motifs within the N-terminal ankyrin repeats of ARMS (ARMS ARs), featuring the conserved W-x-x-L sequence (Figs. 1A and EV1A). To experimentally validate the interaction, we performed co-immunoprecipitation (Co-IP) assays in HEK293T cells, confirming that full-length ARMS interacts with GFP-GABARAP (Fig. 1B). Fast protein liquid chromatography (FPLC) and isothermal titration calorimetry (ITC) assays further verified that purified N-terminal ankyrin repeats of ARMS bind directly to GABARAP (Figs. 1C and EV1B).

To precisely map the binding region, we constructed a series of ARMS truncation variants. GST pull-down assays showed that the 1-4 ankyrin repeats of ARMS (ARMS 1-4 ARs) are responsible for GABARAP binding (Fig. 1D). FPLC and ITC analyses confirmed that purified ARMS 1-4 ARs exhibit high-affinity binding to GABARAP ($K_d$ ~0.9 μM) (Fig. 1E,F). Taken together, these biochemical results identify GABARAP as a novel ARMS interactor and demonstrate that the first four ankyrin repeats of ARMS constitute the primary GABARAP-binding region.

## ARMS specifically interacts with the GABARAP subfamily of the Atg8 family

The Atg8 family in mammals consists of six homologs that can be classified into two subfamilies: the GABARAP subfamily (GABARAP, GABARAPL1, and GABARAPL2) and the LC3 subfamily (LC3A, LC3B, and LC3C). These proteins share high sequence similarity and structural conservation but exhibit distinct functional properties (Wesch et al, 2020). To determine whether ARMS selectively interacts with the GABARAP subfamily or binds to other Atg8 family members, we systematically tested its interactions with all six homologs. FPLC assays demonstrated that ARMS 1-4 ARs bind directly to GABARAP, GABARAPL1, and GABARAPL2 (Figs. 1E and EV1C,D). ITC analysis further revealed that ARMS 1-4 ARs bind GABARAPL1 and GABARAPL2 with comparable affinities (Figs. 1F–H and EV1F). To further validate these interactions in a cellular context, we performed Co-IP assays in HEK293T cells, which confirmed that ARMS co-immunoprecipitates with GABARAP and GABARAPL1, but not with LC3A or LC3B (Fig. EV1E). Finally, FPLC and ITC assays confirmed the absence of detectable interactions between ARMS 1-4 ARs and any LC3 subfamily members (Fig. EV1F–I). Collectively, all these biochemical results demonstrated that ARMS could specifically and discriminately bind to GABARAP subfamily of the Atg8 family.

## Crystal structure of the ARMS-GABARAP complex reveals a non-canonical binding mode

To understand the structural basis of the ARMS-GABARAP interaction, we attempted to determine the complex structure. Following extensive crystallization screening using ARMS truncation constructs and GABARAP fusion proteins, we successfully obtained crystals of ARMS 1-4 ARs fused with GABARAP, which diffracted to 2.0 Å resolution (Appendix Table S1). The structure was solved by molecular replacement, revealing a 1:1 binding stoichiometry between ARMS and GABARAP. In the complex, ARMS adopts a classical ankyrin repeat fold with helix-turn-helix

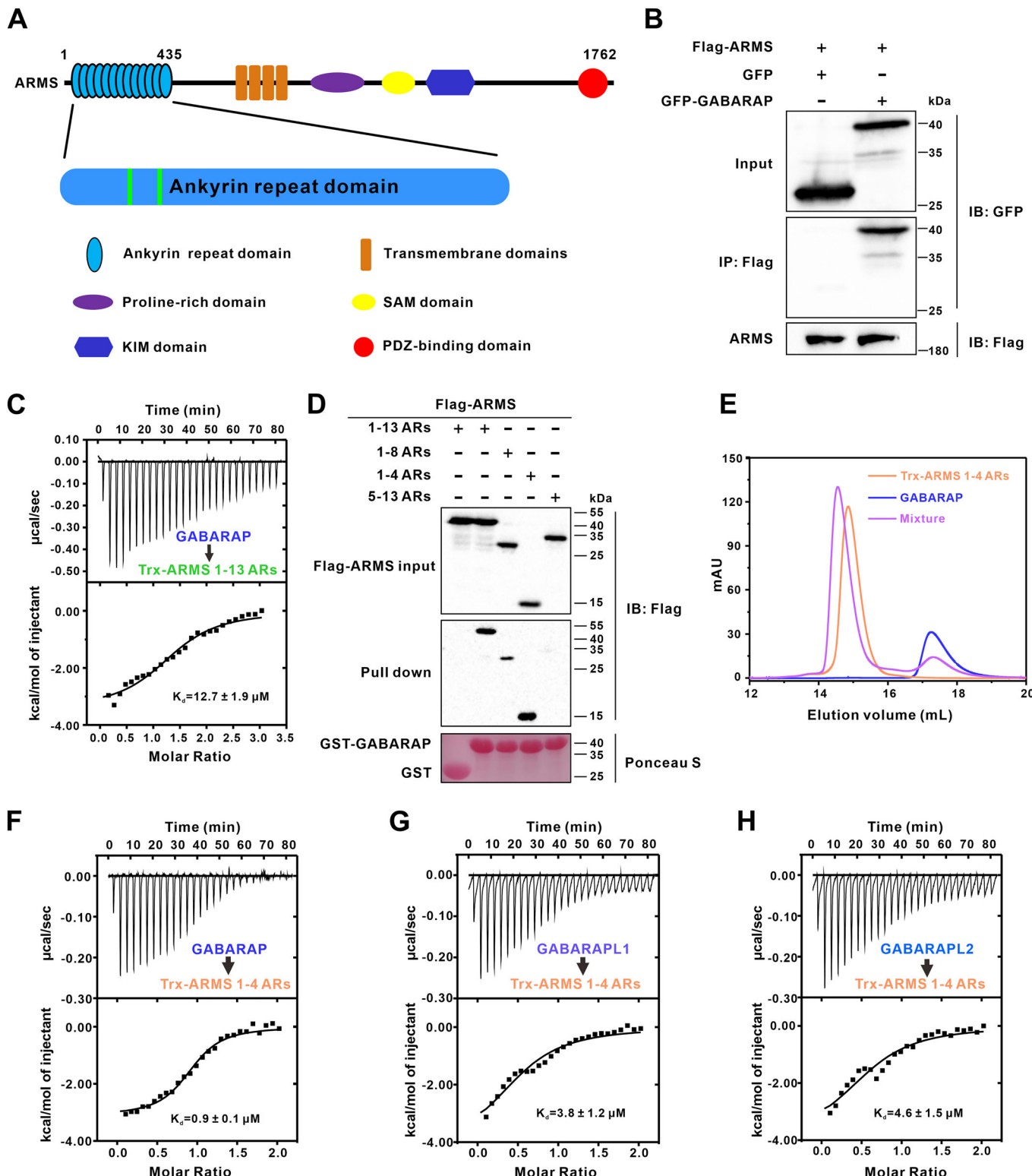

repeats, where finger-like loops (finger loops) connect adjacent repeats (Fig. 2A). GABARAP exhibits the expected ubiquitin-like core fold with two N-terminal α-helices, forming two hydrophobic grooves, HP1 and HP2, which typically function as LIR docking sites (LDS). Surprisingly, ARMS does not interact with GABARAP

via the predicted LIR motif (WTAL), which is located far from the hydrophobic grooves (Fig. 2A,B). Instead, ARMS engages GABARAP through an alternative binding mode, where its first and second finger loops insert into HP2 and HP1, respectively, to facilitate complex formation (Fig. 2A). Unlike canonical LIR-

**Figure 1.  Identification of GABARAP as a novel ARMS interactor.**

(A) Domain architecture of ARMS, highlighting ankyrin repeat domain (ARs), transmembrane domains, and the C-terminal regions. The predicted GABARAP-interacting motif is located within the N-terminal ARs region. (B) Co-immunoprecipitation (Co-IP) assay in HEK293T cells expressing Flag-ARMS and GFP-GABARAP. Anti-Flag beads were used for immunoprecipitation, and western blot analysis confirmed the interaction between Flag-ARMS and GFP-GABARAP. (C) ITC analysis showing that ARMS N-terminal 1–13 ARs directly bind to GABARAP. The dissociation constant ($K_d$) is indicated. (D) In vitro pull-down assay using GST-GABARAP as bait to pull down Flag-tagged ARMS fragments from transfected HEK293T cell lysates. The results identify ARMS 1-4 ARs as the GABARAP-binding region. (E) Fast protein liquid chromatography (FPLC) assay confirming direct interaction between purified Trx-ARMS 1-4 ARs and purified GABARAP. Co-elution of the two proteins indicates complex formation. (F–H) Binding specificity of ARMS to GABARAP subfamily proteins. ITC analysis shows interaction between ARMS 1-4 ARs and GABARAP (F), GABARAPL1 (G), and GABARAPL2 (H), confirming subfamily-specific binding. The $K_d$ values are indicated. Source data are available online for this figure.

dependent interactions, which primarily rely on hydrophobic residues, the ARMS-GABARAP interface is predominantly mediated by charged residues interacting with HP1 and HP2 (Fig. 2B). These findings reveal that ARMS interacts with GABARAP via a previously uncharacterized binding mechanism, independent of the canonical LIR-LDS interaction paradigm.

## Molecular interface of the ARMS-GABARAP complex

The ARMS-GABARAP interaction is primarily stabilized by electrostatic and hydrogen bonding interactions. In the complex structure, the first and second finger loops of ARMS insert into the hydrophobic pockets (HP2 and HP1) of GABARAP, respectively (Fig. 2C,D). At HP1, the negatively charged residues E67 and D70 in the second finger loop (loop2) of ARMS form strong electrostatic interactions with positively charged residues K20, K24, K46, and K48 in GABARAP (Fig. 2C). Similarly, at HP2, the negatively charged residue E36 in the first finger loop (loop1) of ARMS interacts electrostatically with R67 of GABARAP (Fig. 2D). In addition to electrostatic interactions, hydrophobic contacts also contribute to complex stability. The L69 residue in ARMS loop2 inserts into GABARAP HP1, engaging I21 and L50 (Fig. 2E). However, no hydrophobic interactions are observed between ARMS and GABARAP HP2. Instead, L63 of GABARAP interacts with a hydrophobic patch on ARMS, composed of V3, L4, and I5 (Fig. 2F). These structural insights highlight an atypical Atg8 family interaction mode, where ARMS engages GABARAP through a combination of electrostatic interactions and non-canonical hydrophobic contacts, rather than the classical LIR-dependent binding mechanism.

## Characterization of key residues at the ARMS-GABARAP interface

To further validate the interaction mechanism between ARMS and GABARAP, we systematically examined the contributions of key interface residues through site-directed mutagenesis and ITC-based binding assays. Given that the ARMS-GABARAP complex is primarily stabilized by electrostatic and hydrophobic interactions, we first tested mutations targeting electrostatic interactions. The negatively charged residue E67 in ARMS loop2 forms a salt bridge with K20 and K24 in GABARAP HP1 (Fig. 2C). Single mutations K20E or K24E in GABARAP weakened the binding affinity, while the double mutant K20E/K24E and the ARMS E67R mutant completely abolished binding (Fig. 3A,F). Similarly, D70 in ARMS loop2 interacts with K46 and K48 in GABARAP HP1 (Fig. 2C), and mutations K46L or K48E in GABARAP or D70N in ARMS

disrupted the interaction (Fig. 3B,F). In addition, the negatively charged E36 in ARMS loop1 mediates a charge interaction with R67 in GABARAP HP2 (Fig. 2D), and mutations E36R in ARMS or R67E in GABARAP completely eradicated binding (Fig. 3C,F).

In addition to electrostatic interactions, hydrophobic contacts also contribute to complex stability. ITC-based assays showed that mutations L69Q in ARMS or L50A in GABARAP abolished the interaction, while the I21A mutation significantly weakened binding (Fig. 3D,F). Furthermore, the hydrophobic residues V3, L4, and I5 in ARMS interact with L63 of GABARAP, forming a stabilizing hydrophobic patch (Fig. 2F). Although triple-site mutations V3A/L4A/I5A in ARMS only slightly weakened binding, the L63A mutation in GABARAP completely abolished the interaction (Fig. 3E,F). These findings indicate that L63 plays a critical role in maintaining the integrity of GABARAP's hydrophobic groove, facilitating its interaction with ARMS.

To further explore the specificity of this interaction, we performed sequence alignment and structural comparison across the Atg8 family. We found that several key GABARAP residues mediating ARMS binding are absent in LC3 proteins. For instance, the residue corresponding to K20 in GABARAP is an uncharged glutamine in LC3A, a hydrophobic leucine in LC3B, and a neutral glycine in LC3C. Similarly, K24 in GABARAP is replaced by glutamine in LC3A and LC3B (Fig. 3G). Consistent with this, wild-type LC3 proteins fail to bind ARMS. However, introducing K20/K24-equivalent mutations into LC3A, LC3B, and LC3C conferred ARMS-binding capability (Fig. 3H). Notably, the ARMS E67R mutation abolished this newly acquired interaction, confirming the essential role of electrostatic complementarity (Fig. 3H). Collectively, these results establish the molecular determinants of ARMS-GABARAP binding and reveal that electrostatic residues unique to the GABARAP subfamily dictate its interaction specificity. The ability to engineer LC3 proteins to bind ARMS further underscores the distinct biochemical features that differentiate GABARAP from other Atg8 homologs.

## ARMS-GABARAP binding mode differs from canonical Atg8-LIR interactions

Members of the Atg8 protein family typically interact with their binding partners via a conserved LC3-interacting region (LIR) motif, which docks into the hydrophobic grooves HP1 and HP2 of Atg8 proteins. A well-characterized example is the GABARAPL1-ATG14 interaction, where the LIR motif in ATG14 (W435 and L438) inserts into HP1 and HP2 of GABARAPL1, forming a stable hydrophobic interface (Fig. EV2A,B). Similarly, UBA5 engages GABARAP through a hydrophobic W431 residue that

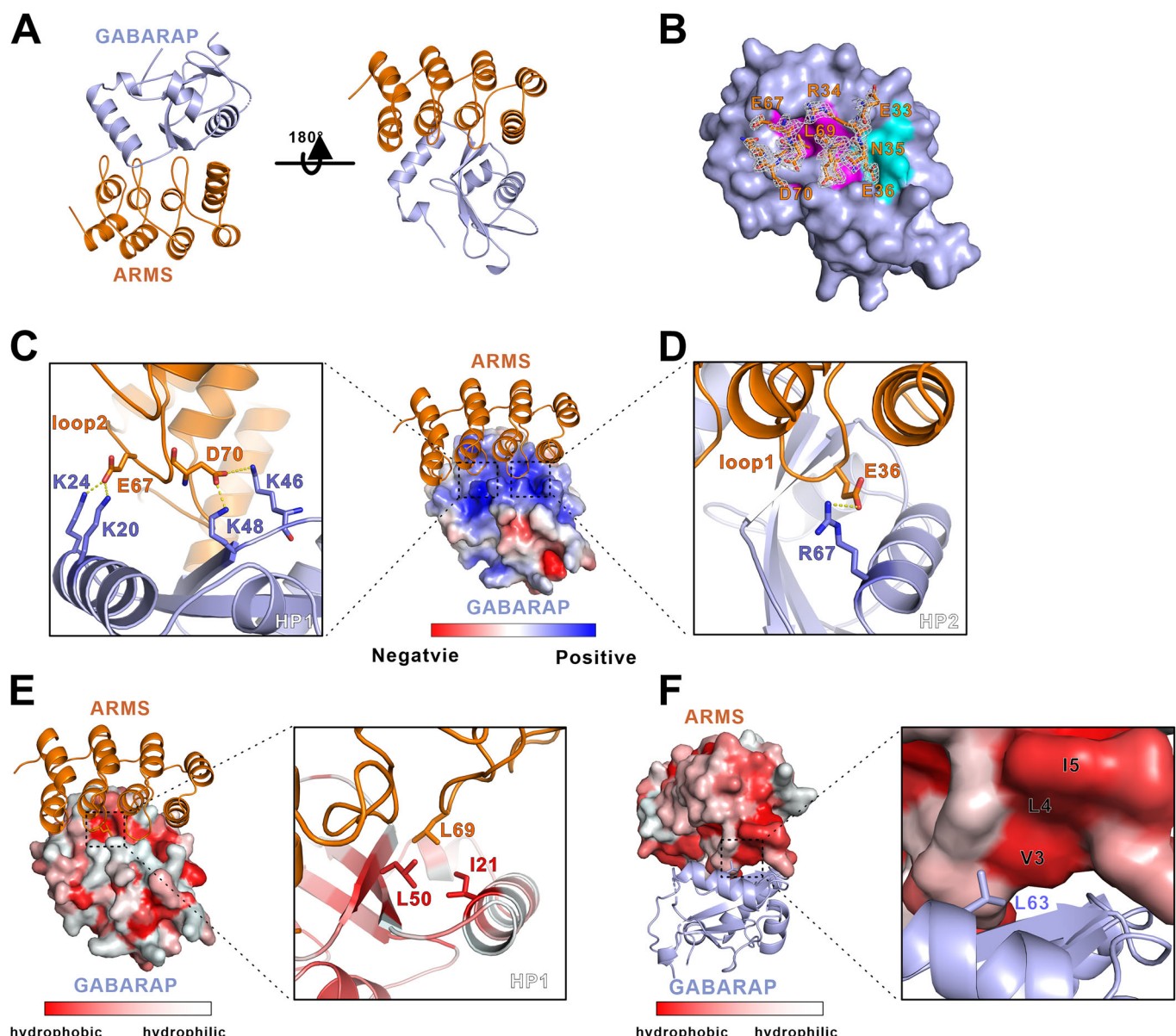

**Figure 2. Overall structure of ARMS-GABARAP complex and interaction interface.**

(A) Ribbon representation of the ARMS 1-4 ARs-GABARAP complex. ARMS 1-4 ARs is depicted in orange, and GABARAP is shown in light blue. The two loops connecting ARMS ankyrin repeats insert into the hydrophobic pockets of GABARAP. (B) Electron density map (Fo-Fc) of the ARMS-GABARAP complex, demonstrating that key ARMS residues can be unambiguously assigned. ARMS is shown as sticks, with critical residues E33, R34, N35, E36, E67, L69, and D70 highlighted. GABARAP hydrophobic pocket 1 (HP1) is colored pink, and hydrophobic pocket 2 (HP2) is colored cyan. (C) Electrostatic interactions within HP1. The E67 and D70 residues in loop2 of ARMS form charge interactions with positively charged residues K20, K24, K46, and K48 of GABARAP HP1. (D) Electrostatic interactions within HP2. The E36 residue in loop1 of ARMS interacts with positively charged residue R67 in GABARAP HP2, stabilizing the complex. (E) Hydrophobic interactions in HP1 of GABARAP. The L69 residue in loop2 of ARMS inserts into HP1 of GABARAP, consisting of I21, L50, and other hydrophobic residues. (F) Hydrophobic interactions within the ARMS binding surface. The hydrophobic interface formed by ARMS residues (V3, L4, and I5) enhances complex formation with GABARAP. The critical residue L63 of GABARAP interacts with the hydrophobic interface of ARMS. Source data are available online for this figure.

inserts into an additional binding site, HP0, while its hydrophobic residues I343, L345, and V346 interact with HP1 and HP2 (Fig. EV2C,D). In contrast, the ARMS-GABARAP interaction deviates from this canonical LIR-LDS binding paradigm. Although ARMS binds to GABARAP at HP1 and HP2, it does not utilize a typical LIR motif (Fig. EV2E). Instead, ARMS engages HP1 via a combination of hydrophobic and electrostatic interactions, while HP2 binding is exclusively mediated by electrostatic contacts (Fig. EV2F). Notably, the hydrophobic residue L63 of GABARAP stabilizes the interaction by engaging a hydrophobic surface on ARMS, further reinforcing this non-canonical binding mode (Fig. 2F). Together, these structural and biochemical findings

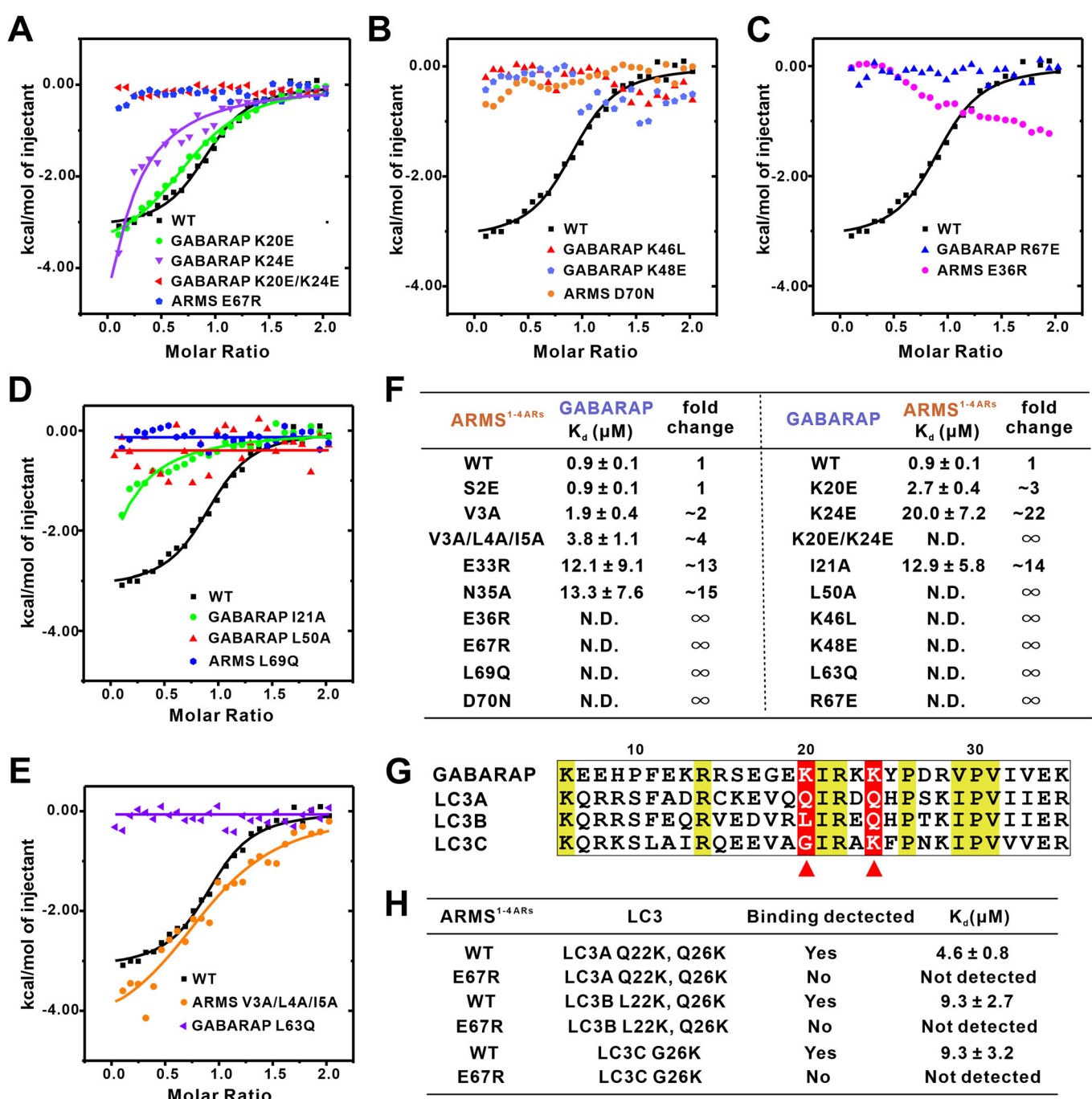

demonstrate that ARMS employs a distinct mode of interaction with GABARAP, diverging from the classical LIR-mediated docking strategy.

## ARMS is required for dendritic and synaptic development in hippocampal neurons

As a critical scaffold protein, ARMS is involved in essential neuronal processes, including survival, growth, and signal transmission (Cesca et al, 2011; Scholz-Starke and Cesca, 2016). To investigate its functional role in neurons, we first analyzed its endogenous distribution across different developmental stages. Immunostaining revealed that ARMS is consistently expressed throughout neuronal growth, localizing to the soma, dendrites, and axons, with prominent accumulation near the Golgi apparatus (Fig. EV3A–F). This suggests that ARMS may participate in protein trafficking, modification, or transport, processes crucial for neuronal development. To determine whether ARMS and GABARAP interact in neurons, we performed co-immunoprecipitation (Co-IP) in primary hippocampal cultures and observed a robust endogenous interaction between ARMS and GABARAP (Fig. EV3G), supporting their physiological association.

**Figure 3.  Structure-guided mutational analysis of the ARMS-GABARAP interaction.**

(**A, B**) Electrostatic interactions in HP1. GABARAP HP1 contains positively charged residues (K20, K24, K46, K48), which interact with negatively charged ARMS residues (E67 and D70). Mutations E67R or D70N (ARMS) and K20E/K24E, K46L, K48E (GABARAP) disrupt the interaction between ARMS and GABARAP. Single mutation K20E or K24E of GABARAP impair binding with ARMS. (**C**) Electrostatic interactions in HP2. GABARAP HP2 R67 interacts with ARMS E36, forming additional charge complementarity. Mutations E36R (ARMS) or R67E (GABARAP) abolish the interaction. (**D**) Hydrophobic interactions in HP1. Mutations L69Q (ARMS) and L50A (GABARAP) abolish the interaction, while I21A reduced the interaction. (**E**) Hydrophobic interactions at the ARMS ankyrin repeat surface. The first ankyrin repeat of ARMS (V3, L4, I5) forms a hydrophobic patch that interacts with GABARAP L63. Mutation of L63Q (GABARAP) abolishes the interaction, while V3A/L4A/I5A (ARMS) reduced the interaction. (**F**) ITC-derived binding affinities summarizing the effects of mutations on ARMS-GABARAP interaction. Dissociation constants ($K_d$) were measured to compare the binding affinities of mutant ARMS 1-4 ARs or mutant GABARAP with their respective wild-type counterparts. The results demonstrate that electrostatic and hydrophobic interactions play a critical role in stabilizing the ARMS-GABARAP complex, as mutations at key interface residues significantly weaken binding. (**G**) Sequence alignment of mouse GABARAP (6-35), LC3A (8-37), LC3B (8-37), and LC3C (14-43). Conserved residues are highlighted in yellow boxes. The K20 and K24 in GABARAP are not conserved in the LC3 subfamily (red arrows). (**H**) ITC-derived binding affinities summarizing the effects of mutations on ARMS-LC3 interaction. LC3A (Q22K/Q26K), LC3B(L22K/Q26K), and LC3C (G26K) enables detectable binding to ARMS 1-4 ARs, The E67R mutant of ARMS disrupts the interaction. Source data are available online for this figure.

To dissect the molecular basis of ARMS–GABARAP binding, we introduced interface-disrupting mutations in ARMS and evaluated their impact in HEK293T cells. Co-IP assays confirmed that wild-type ARMS strongly associates with GABARAP, whereas single mutations E33R and L69Q reduced the interaction. Notably, both the E33R/E67R (EE) and E33R/L69Q double mutations completely abolished GABARAP binding (Fig. 4A). To verify that these mutations do not compromise the structural integrity of ARMS, we performed circular dichroism (CD) spectroscopy, which showed no detectable alterations in secondary structure for any of the mutants (Fig. 4B). Additionally, immunoblotting confirmed that wild-type and mutant ARMS proteins were expressed at comparable levels in cells (Fig. EV4A,B). These results indicate that the observed changes in GABARAP binding are due to specific disruption of the interface, rather than global misfolding or expression differences.

To functionally evaluate the role of ARMS during neuronal development, we performed knockdown and rescue experiments in cultured hippocampal neurons. ARMS depletion using ARMS-shRNA at day 4 significantly reduced total dendritic length by day 7, compared to neurons transfected with the control vector (Fig. 4C,D). Expression of wild-type ARMS restored dendritic growth, whereas the GABARAP-binding deficient mutant (ARMS EE, E33R/E67R) exhibited a weaker rescue effect (Fig. 4C,D). These findings indicate that ARMS is essential for dendritic growth, and its function is at least partially dependent on its interaction with GABARAP. During synaptogenesis (day 10–14), ARMS knockdown led to a significant reduction in dendritic spine density, demonstrating its essential role in synapse formation (Fig. 4E,F). Although both wild-type ARMS and ARMS EE partially restored spine density, the mutant was less effective (Fig. 4E,F). These results suggest that ARMS is a critical regulator of synaptic development, and its interaction with GABARAP may be involved in the process of synaptogenesis.

## GABARAP and ATG7 are required for ARMS-dependent dendritic and synaptic development

To further investigate whether GABARAP is functionally required for ARMS-mediated neuronal development, we co-transfected hippocampal neurons with ARMS-targeting shRNA and a rescue construct encoding either wild-type ARMS (WT) or the GABARAP-binding–deficient mutant ARMS EE, in the presence or absence of GABARAP knockdown. At day 4 of neuronal culture,

simultaneous depletion of ARMS and GABARAP impaired dendritic growth, but overexpression of ARMS WT largely restored dendritic complexity. In contrast, the ARMS EE mutant failed to rescue the phenotype, resulting in persistently reduced dendritic arborization (Fig. EV3H,I). These results suggest that GABARAP is required for ARMS to support early dendritic development. We next extended the analysis to later developmental stages. At day 10, neurons were co-transfected as above and analyzed on day 14. Notably, both ARMS WT and EE failed to rescue spine density under GABARAP-depleted conditions, further supporting that GABARAP is essential for ARMS-mediated regulation of both dendritic and synaptic development (Fig. EV3J,K).

To determine whether this regulation involves the autophagy pathway, we knocked down ATG7, a key autophagy-related gene, in parallel with ARMS rescue. ARMS WT successfully restored dendritic growth at day 7 under ATG7 knockdown, but failed to rescue spine defects at day 14, closely mimicking the phenotype observed under GABARAP depletion. In contrast, ARMS EE was unable to rescue either dendritic or synaptic defects under ATG7 knockdown conditions, further supporting the notion that ARMS requires autophagy machinery for its full functionality (Fig. EV3H–K). These results suggest that autophagy is necessary for ARMS-mediated neuronal maturation, and that both GABARAP and ATG7 are required for its full function during dendritic and synaptic development.

## GABARAP modulates ARMS-mediated dendritic spine growth and maturation

To investigate whether the ARMS-GABARAP complex plays a role in dendritic spine development, we overexpressed RFP-tagged wild-type ARMS (ARMS WT) or the GABARAP-binding–deficient mutant (ARMS EE) in hippocampal neurons at day 10 and performed immunofluorescence analysis at day 18. Before assessing their functional effects, we first ruled out the possibility that ARMS overexpression leads to non-specific aggregation. Triton solubility assays in HEK293T cells showed that both endogenous and overexpressed Flag-ARMS remained predominantly soluble, with only minimal protein detected in the pellet fraction (Fig. EV4C,D). Even under chloroquine (CQ)-induced autophagy inhibition, only trace amounts of ARMS shifted to the insoluble fraction in neurons (Fig. EV4E), indicating that ARMS overexpression does not cause protein aggregation or non-physiological interactions.

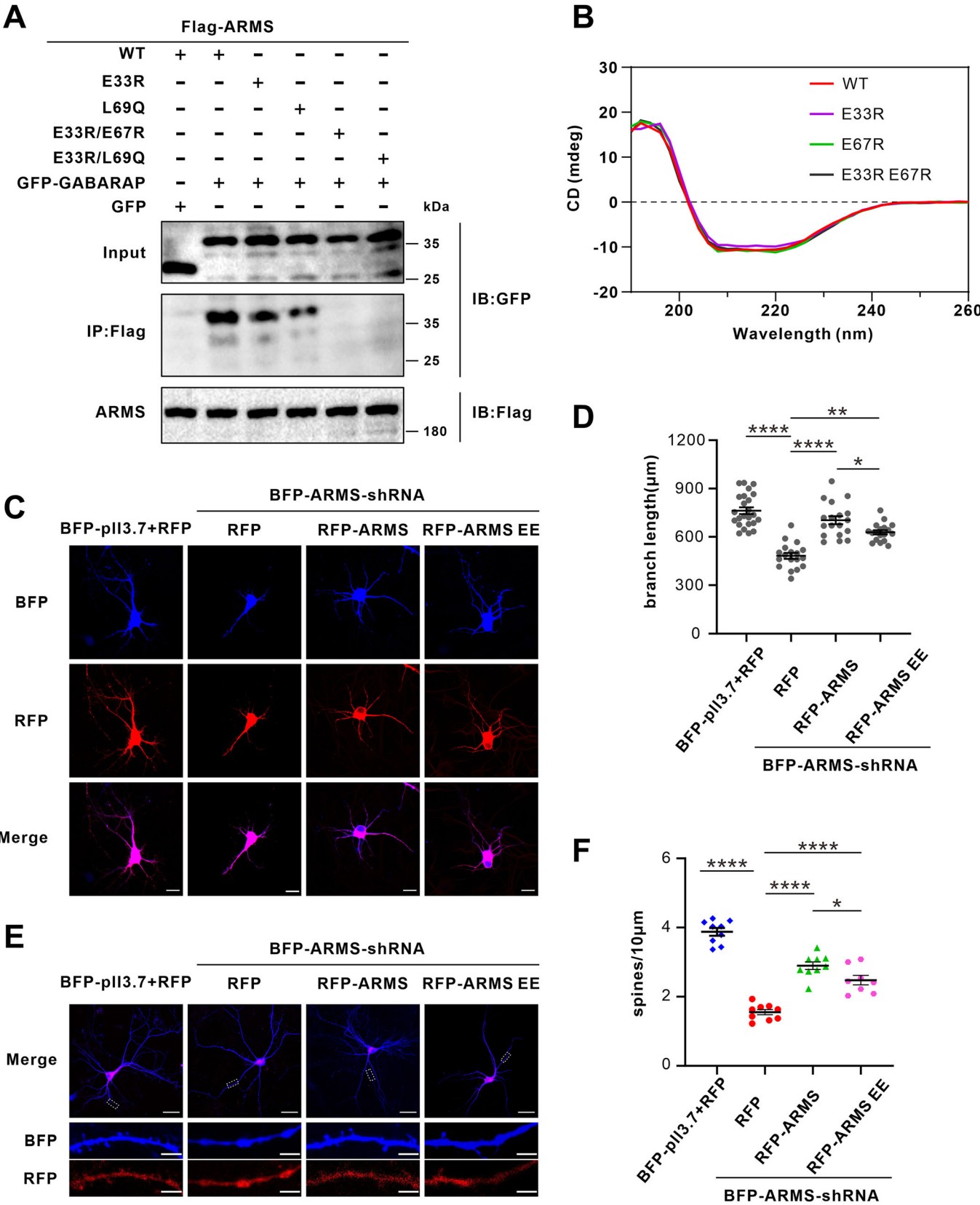

**Figure 4.  ARMS promotes dendrite and spine growth in hippocampal neurons.**

(A) Co-IP assay showing the interaction between Flag-ARMS WT or mutants with GFP-GABARAP in HEK293T cells. Mutations E33R/E67R (EE, hereafter) or E33R/L69Q of ARMS disrupt ARMS-GABARAP binding in cells. (B) Purified Trx-ARMS 1–4ARs (WT), E33R, E67R, and E33R/E67R double mutant proteins were analyzed by circular dichroism, showing that the mutations do not perturb the secondary structure of the ankyrin-repeat domain. (C) Hippocampal neurons were transfected at day 4 with BFP-pll3.7 and RFP, BFP-ARMS-shRNA and RFP, BFP-ARMS-shRNA and RFP-ARMS, or BFP-ARMS-shRNA and RFP-ARMS EE (E33R/E67R mutant, RFP-tagged). The neurons were imaged under confocal microscope and representative images are shown; scale bar, 20 μm. (D) Quantification of neurite length in transfected neurons (BFP-pll3.7 and RFP, $n = 24$; BFP-ARMS-shRNA and RFP, $n = 18$; BFP-ARMS-shRNA and RFP-ARMS, $n = 18$; BFP-ARMS-shRNA and RFP-ARMS EE, $n = 18$). Error bars represent mean ± SEM. Data were normally distributed, and one-way ANOVA was used for statistical analysis. ****$P < 0.0001$, **$P = 0.0039$, *$P = 0.0109$. (E) Dendritic spine analysis in hippocampal neurons. Neurons were transfected at day 10 with BFP- pll3.7 or BFP-ARMS-shRNA, rescued with RFP, RFP-ARMS, or RFP-ARMS EE, and imaged at day 14. Upper panels: low-magnification images, white box highlights the region of interest (scale bar, 50 μm). Lower panels: high-magnification images showing dendritic spines (scale bar, 5 μm). (F) Quantification of dendritic spine density from three independent hippocampal cultures (BFP-pll3.7 and RFP, BFP-ARMS-shRNA and RFP, BFP-ARMS-shRNA and RFP-ARMS, $n = 9$; BFP-ARMS-shRNA and RFP-ARMS EE, $n = 8$). Error bars represent mean ± SEM. Data were normally distributed, and one-way ANOVA was used for statistical analysis. ****$P < 0.0001$, *$P = 0.0333$. Source data are available online for this figure.

Overexpression of both ARMS WT and ARMS EE significantly increased dendritic spine density compared to control neurons; however, the effect of ARMS EE was markedly weaker (Fig. 5A,B). Notably, ARMS WT also promoted the formation of mature, mushroom-shaped spines, while ARMS EE failed to induce spine maturation (Fig. 5C), suggesting that GABARAP binding is critical for ARMS-mediated spine morphogenesis.

To further assess the role of GABARAP, we co-expressed GFP-GABARAP or its mutants with ARMS in neurons. Immunoblotting confirmed that GFP-GABARAP and its variants were expressed at comparable levels (Fig. EV4F,G). Co-expression of GABARAP WT significantly reduced dendritic spine density and impaired spine maturation, compared to ARMS WT alone (Fig. 5D–F), indicating that GABARAP negatively regulates ARMS function in spine development. In contrast, co-expression of GABARAP with the ARMS EE alter dendritic spine number, but did not promote spine maturation, suggesting that GABARAP's inhibitory effect requires direct binding to ARMS (Fig. 5D–F).

To dissect the functional specificity of this interaction, we utilized two GABARAP mutants: L63Q, which is deficient in ARMS binding, and G116 A, which cannot undergo lipidation. Co-expression of ARMS WT with GABARAP L63Q resulted in a partial suppression of spine density and full rescue of spine maturation, compared to co-expression with GABARAP WT. This suggests that GABARAP's inhibitory effect on spine number is partly dependent on its interaction with ARMS, while inhibition of spine maturation requires direct ARMS binding. In contrast, the lipidation-deficient mutant GABARAP G116A strongly suppressed both spine density and maturation, indicating that GABARAP's autophagic activity also contributes to modulating ARMS function (Fig. 5D–F). Together, these findings demonstrate that ARMS promotes dendritic spine growth and maturation, and that GABARAP modulates this activity via both direct interaction and its role in autophagy. The inability of the ARMS EE mutant to induce spine maturation underscores the functional relevance of GABARAP binding in regulating ARMS-mediated spine development.

## GABARAP regulates ARMS protein levels through an autophagy-dependent mechanism

To investigate how GABARAP regulates ARMS abundance, we first examined its effect on both endogenous and overexpressed ARMS in HEK293T cells. Expression of GFP-GABARAP or the ARMS-

binding deficient mutant GFP-GABARAP L63Q had no detectable impact on endogenous ARMS protein levels (Fig. EV5A,B). However, upon overexpression of Flag-ARMS, wild-type GABARAP significantly reduced ARMS levels, whereas the L63Q mutant or LC3B failed to do so (Fig. 6A,B), indicating that GABARAP selectively targets excess ARMS for degradation through direct binding.

Given that GABARAP is a key regulator of autophagy (Schaaf et al, 2016; Birgisdottir et al, 2013), we next investigated whether autophagy influences ARMS stability. Inhibition of autophagic flux with chloroquine (CQ) led to a marked accumulation of endogenous ARMS (Fig. 6C,D), suggesting that ARMS is subject to autophagy-mediated degradation. Consistently, knockdown of GABARAP or the core autophagy factor ATG7 in HEK293T cells also elevated ARMS protein levels (Figs. 6E,F and  EV5C,D). Similarly, treatment of cultured neurons with an ATG7 inhibitor (ATG7-IN-2) increased ARMS levels (Fig. EV5E,F), supporting the notion that ARMS degradation is coupled to autophagy in both cell types. To test whether GABARAP-mediated ARMS degradation requires intact autophagy, we overexpressed ARMS with or without GABARAP in the presence of CQ. Inhibition of autophagy partially rescued the GABARAP-induced reduction in ARMS levels (Fig. 6G,H), confirming that GABARAP promotes ARMS turnover through an autophagy-dependent mechanism.

We next asked whether ARMS is physically routed through the autophagic machinery. We found that 3 h of EBSS induction led to significant LC3B aggregation in cells (Fig. EV5G). Upon EBSS-induced autophagy, cells overexpressing RFP-ARMS exhibited strong upregulation of the lysosomal marker LAMP1 and extensive co-localization between ARMS and LAMP1 (Fig. EV5H), suggesting lysosomal accumulation. Unlike typical LAMP1 puncta, we observed extended or irregular LAMP1 structures, possibly reflecting high autophagic flux due to combined autophagy induction and ARMS overexpression. Co-expression of GFP-LC3B with RFP-ARMS revealed diffuse LC3B distribution under basal conditions, which shifted to prominent puncta colocalizing with ARMS following EBSS treatment (Fig. EV5I). Furthermore, a protease protection assay demonstrated that Flag-ARMS was shielded from degradation unless detergent was applied, indicating its sequestration within autophagosomal membranes (Fig. 6I).

Together, these findings identify ARMS as a bona fide substrate of autophagic degradation. GABARAP serves as a selective adaptor that facilitates the turnover of excess ARMS by promoting its targeting to autophagosomes.

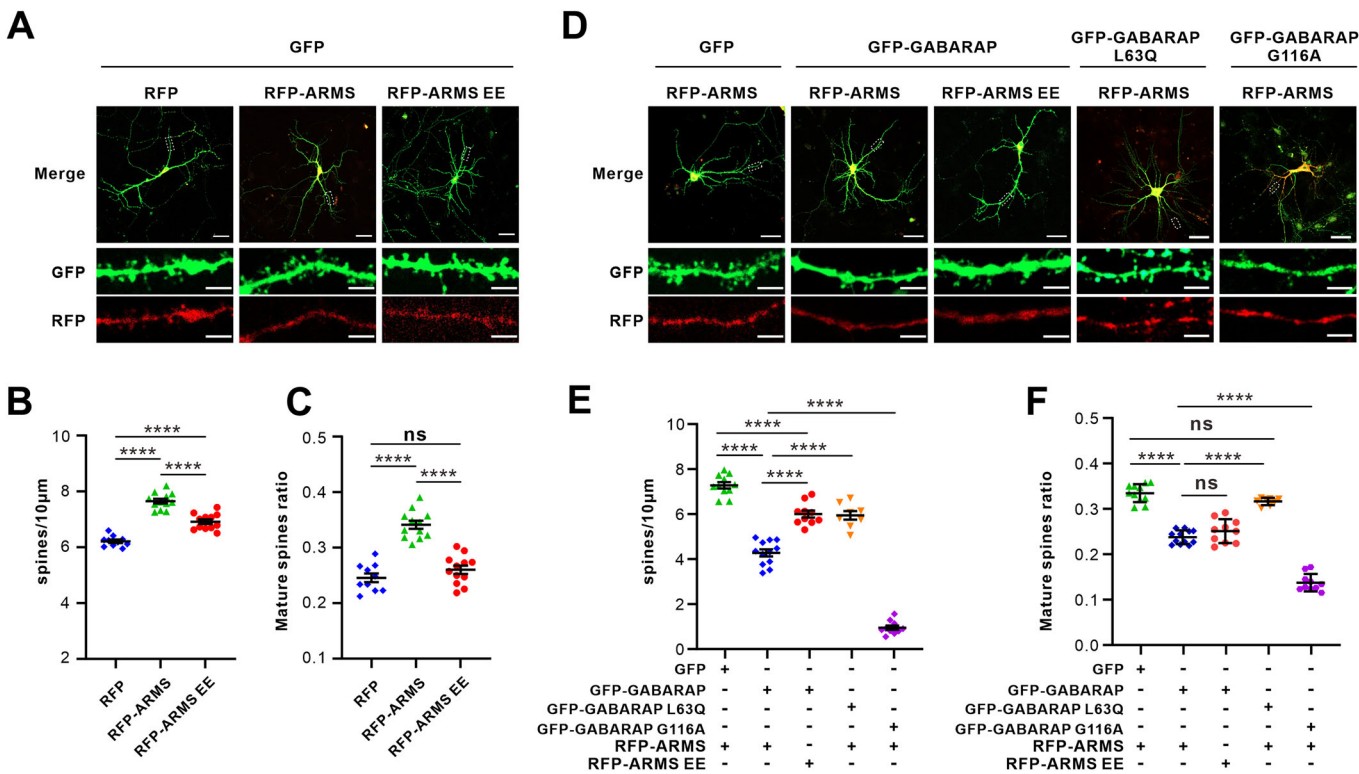

**Figure 5. The ARMS-GABARAP complex regulates synaptic development.**

(A) Hippocampal neurons were transfected at day 10 with RFP, RFP-ARMS, or RFP-ARMS EE, followed by immunostaining at day 18. GFP marks transfected neurons (green) co-expressing RFP-ARMS or RFP-ARMS EE. Upper panels: low-magnification images, white box highlights the region of interest (scale bar, 50 μm). Lower panels: high-magnification images of dendritic spines (scale bar, 5 μm). (B, C) Quantification of dendritic spine density (B) and mature spine ratio (C). Spine density was quantified in neurons expressing RFP, RFP-ARMS, or RFP-ARMS EE from three independent hippocampal cultures. Error bars represent mean ± SEM. Data were normally distributed, and one-way ANOVA was used for statistical analysis. ****P < 0.0001. The proportion of mature mushroom-shaped spines was quantified (GFP and RFP, n = 10; GFP and RFP-ARMS, n = 12; GFP and RFP-ARMS EE, n = 12). Error bars represent mean ± SEM. Data were normally distributed, and one-way ANOVA was used for statistical analysis. ****P < 0.0001; ns, not significant. (D) Hippocampal neurons were co-transfected at day 10 with the indicated constructs, and immunostaining at day 18, the white box is region of interest (upper panels, scale bar = 50 μm). Dendrite spines images are shown at high magnification (lower panels, scale bar = 5 μm). (E, F) Quantification of dendrite spines numbers from three independent hippocampal cultures (E) and the ratio of mature mushroom-shape spines (F). All data were measured from three independent hippocampal cultures (GFP and RFP-ARMS, n = 11; GFP-GABARAP and RFP-ARMS, n = 12; GFP-GABARAP and RFP-ARMS EE, n = 10, GFP-GABARAP L63Q and RFP-ARMS, n = 8; GFP-GABARAP G116A and RFP-ARMS, n = 10). Error bars represent mean ± SEM. Data were normally distributed, and one-way ANOVA was used for statistical analysis. ****P < 0.0001; ns not significant. Source data are available online for this figure.

## Ankyrin-derived peptides disrupt the ARMS-GABARAP complex and alter ARMS subcellular localization in neurons

To test whether disrupting ARMS-GABARAP interaction affects ARMS stability, we employed ankyrin-derived peptides that selectively bind GABARAP and block its interactions. Previous studies have shown that LIR-based peptides exhibit moderate affinities for Atg8 family proteins, whereas ankyrin-derived peptides (e.g., AnkB WT) display high-affinity binding to GABARAP, thereby disrupting Atg8-mediated interactions (Li et al, 2018; Chan et al, 1993) (Fig. 7A). Consistent with these findings, ITC assays confirmed that AnkB WT peptide binds GABARAP with high affinity, while AnkB WR peptide, a negative control, does not interact with GABARAP (Fig. 7B). To determine whether AnkB WT competes with ARMS for GABARAP binding, we performed competition ITC assays. When AnkB WT peptide was titrated into a pre-formed GABARAP–ARMS complex, strong

binding was observed, indicating that AnkB WT can still bind GABARAP in the presence of ARMS. In contrast, when ARMS was titrated into a pre-formed GABARAP-AnkB WT complex, no binding signal was detected, suggesting that AnkB WT outcompetes ARMS for GABARAP binding (Fig. 7C). Further FPLC analysis showed that both ARMS and AnkB WT individually associate with GABARAP but do not form a ternary complex (Fig. 7D). Additionally, GST pull-down assays demonstrated that AnkB WT disrupts ARMS-GABARAP complex formation, while AnkB WR has no such effect (Fig. 7E).

To assess the functional consequence of disrupting the ARMS–GABARAP interaction in neurons, we transfected cultured hippocampal neurons at DIV4 with GFP, GFP-AnkB WT, or GFP-AnkB WR, and analyzed ARMS localization at DIV7. Immunoblotting confirmed that AnkB WT and WR peptides are expressed at comparable levels in neurons, ruling out expression-level artifacts (Fig. EV4H–K). Immunostaining for endogenous ARMS revealed that AnkB WT peptide expression markedly increased

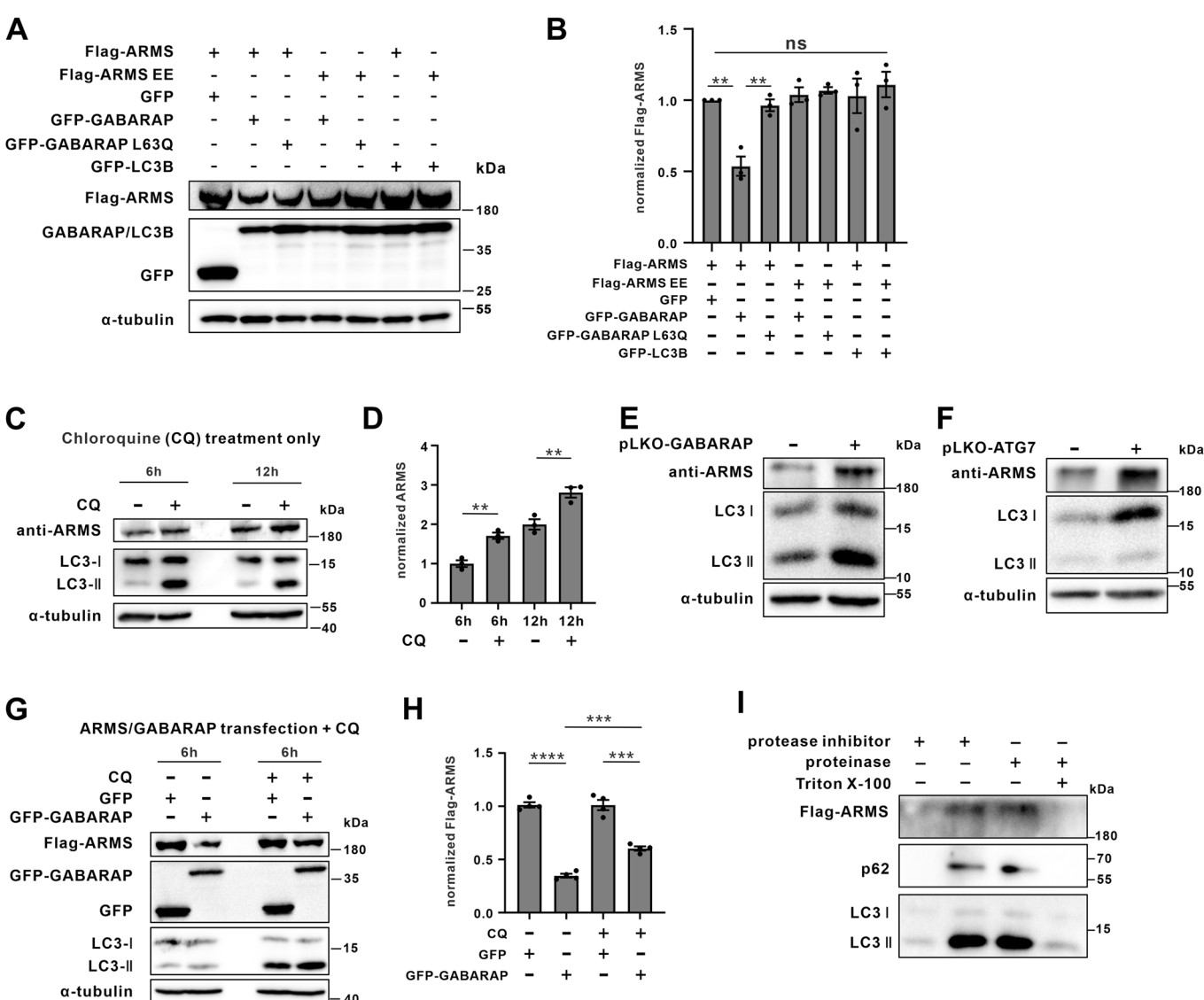

**Figure 6. GABARAP inhibits the abnormal accumulation of ARMS in cells.**

(A) HEK293T cells were co-transfected with Flag-ARMS WT or Flag-ARMS EE together with GFP, GFP-GABARAP, GFP-GABARAP L63Q, or GFP-LC3B as indicated. Western blot analysis showed that co-expression of GABARAP markedly reduced ARMS WT protein levels, whereas the L63Q mutant and LC3B had no significant effect. Notably, GABARAP did not reduce the levels of the ARMS EE mutant. (B) Quantification of Flag-ARMS band intensity normalized to GFP from three independent experiments. Data represent mean ± SEM. Data were normally distributed, and one-way ANOVA was used for statistical analysis. **P < 0.01; ns, not significant. (C, D) CQ treatment increases endogenous ARMS levels in HEK293T cells. Cells were treated with CQ for 6 or 12 h without prior transfection, and western blot analysis of LC3 and ARMS was performed. LC3-II accumulation confirms effective autophagy inhibition (C). Quantification of ARMS levels from three independent experiments (n = 3). Error bars represent mean ± SEM. Data were normally distributed, and one-way ANOVA was used for statistical analysis. **P < 0.01 (D). (E) Lentiviral shRNA-mediated knockdown of GABARAP was performed in HEK293T cells. Western blot analysis revealed a marked increase in endogenous ARMS protein compared to control. (F) Knockdown of ATG7 was similarly achieved via lentiviral shRNA. Western blot analysis showed a robust increase in ARMS protein levels. (G, H) CQ rescues GABARAP-mediated suppression of ARMS accumulation. HEK293T cells were transfected with Flag-ARMS and GFP-GABARAP or GFP, followed by CQ or DMSO treatment for 6 h. Western blot analysis shows that CQ prevents GABARAP-mediated ARMS degradation, leading to ARMS accumulation. Quantification of ARMS levels from four independent experiments (n = 4). Data are presented as mean ± SEM. Data were normally distributed, and one-way ANOVA was used for statistical analysis. ****P < 0.0001, ***P < 0.001 (H). (I) Protease protection assays in HEK293T cells. Transfected HEK293T cells with Flag-ARMS, after 30 h, cells were treated with chloroquine for 10 h. Following cell lysis, autophagosomes were collected via gradient centrifugation, treated with reagents as indicated, and subjected to western blot (WB) analysis. Source data are available online for this figure.

ARMS accumulation in the soma, whereas AnkB WR had no effect (Fig. 7F). Quantification of fluorescence intensity confirmed that AnkB WT peptide promotes ARMS accumulation in the soma of hippocampal neurons (Fig. 7G).

These findings demonstrate that ankyrin-derived peptides effectively disrupt the ARMS-GABARAP complex, leading to altered ARMS subcellular localization and accumulation in the neuronal soma.

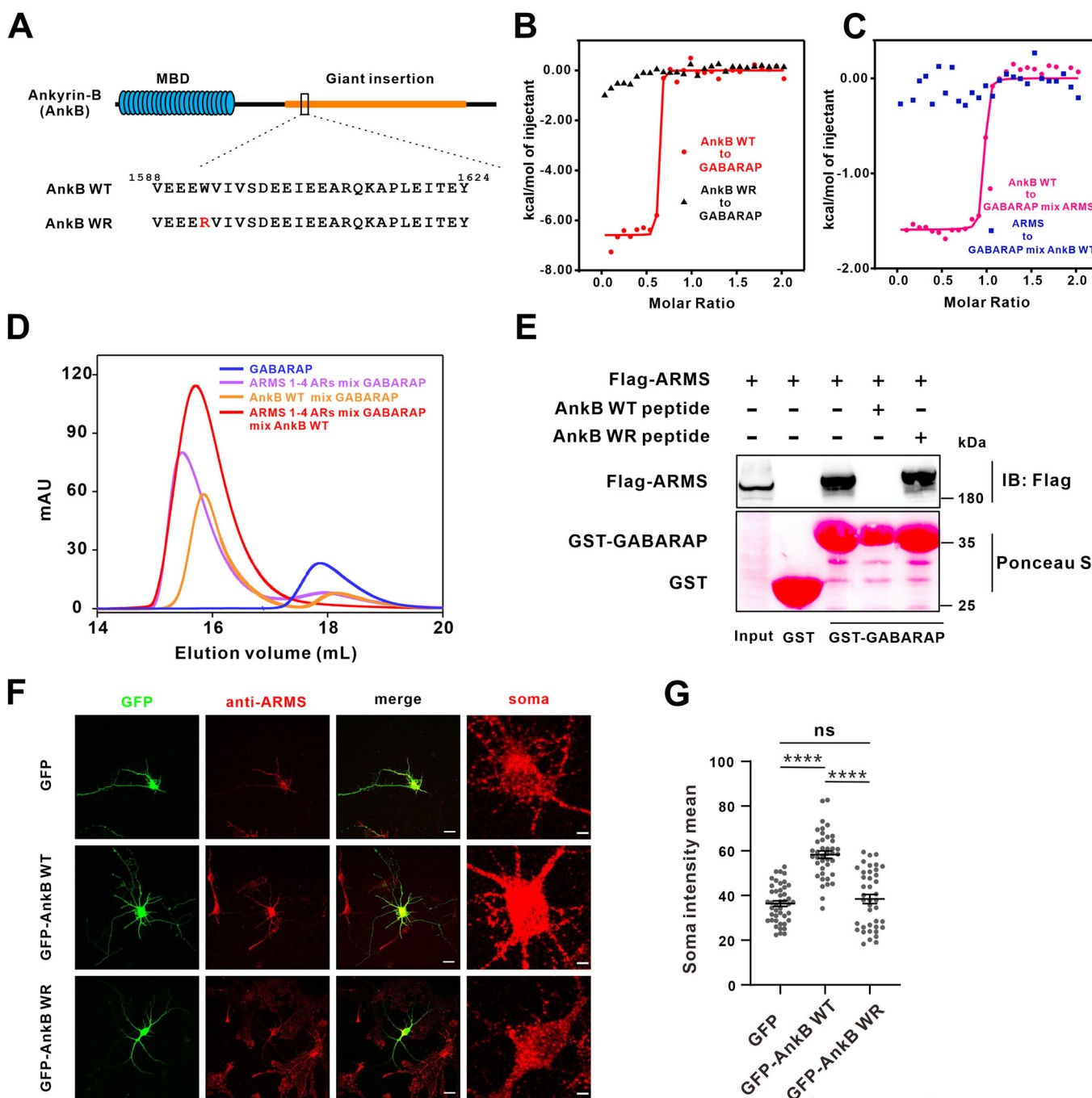

## Discussion

Scaffold proteins are crucial organizers of neuronal signaling, cytoskeletal dynamics, and synaptic structure, serving as integration hubs for diverse pathways involved in development and plasticity (Scholz-Starke and Cesca, 2016). ARMS was initially identified as a crucial neuronal scaffold that integrates neurotrophic signaling and regulates synaptic stability, dendritic arborization, and neuronal survival (Wu et al, 2009; Riol-Blanco et al, 2004). Notably, ARMS also plays a role in various non-neuronal processes, including tumor cell proliferation and immune cell activation (Herr et al, 2024), as alterations in ARMS expression have been linked to malignancies such as melanoma (Liao et al, 2011),

neuroblastoma (Rogers and Schor, 2013) and ovarian cancer (Raza et al, 2018; Bondeson et al, 2022), raising the possibility that dysregulated ARMS levels may perturb signaling homeostasis or lead to aberrant protein clustering. Thus, understanding how ARMS levels and localization are regulated is important not only for neuronal function but also for broader aspects of cellular physiology and disease.

Here, we identify GABARAP as a previously unrecognized ARMS-binding partner that regulates ARMS stability, localization, and function in neurons. Structural and biochemical analyses demonstrate that ARMS binds GABARAP via a non-canonical mode distinct from the classical LIR-LDS interaction (Rogov et al, 2023). In the crystal structure, the finger loops of ARMS insert into

**Figure 7.  Ankyrin-derived peptides disrupt ARMS-GABARAP interaction and increase ARMS levels in the soma of hippocampal neurons.**

(**A**) Schematic representation of the domain organization of 440 kDa Ankyrin-B. (**B**) ITC analysis of AnkB-derived peptides binding to GABARAP. ITC results show that AnkB WT peptide binds GABARAP with high affinity, whereas AnkB WR peptide does not interact with GABARAP, which serves as a negative control. (**C**) Competitive ITC analysis of AnkB WT, ARMS, and GABARAP interactions. When AnkB WT peptide was titrated into a pre-formed ARMS-GABARAP complex, strong binding was still observed, indicating that AnkB WT can interact with GABARAP in the presence of ARMS. In contrast, when ARMS was titrated into a pre-formed AnkB WT-GABARAP complex, no binding was detected, suggesting that AnkB WT binds GABARAP with higher affinity and competitively displaces ARMS. (**D**) FPLC analysis showing that AnkB WT peptide competes with GABARAP for binding to ARMS 1-4 ARs, supporting its role in disrupting ARMS-GABARAP complex formation. (**E**) GST pull-down assays confirming that AnkB WT peptide disrupts the interaction between ARMS and GABARAP. (**F**) Hippocampal neurons were transfected at day 4 with GFP, GFP-AnkB WT peptide, or GFP-AnkB WR peptide, fixed at day 7, and immunostained for endogenous ARMS (red). Z-stack projection images were acquired using confocal microscopy (scale bar, 50 μm). The last column shows a magnified view of the soma region (scale bar, 10 μm), showing AnkB WT peptide increases ARMS intensity in the soma of hippocampal neurons. (**G**) Quantification of normalized ARMS fluorescence intensity in the soma of transfected neurons. The fluorescence intensity was measured in neurons expressing GFP ($n = 44$), GFP-AnkB WT peptide ($n = 40$), and GFP-AnkB WR peptide ($n = 40$). AnkB WT peptide significantly increases ARMS intensity in the soma. Error bars indicate SEM. Data were normally distributed, and one-way ANOVA was used for statistical analysis. ****$P < 0.0001$, ns, not significant. Source data are available online for this figure.

the HP1 and HP2 pockets of GABARAP through a combination of electrostatic and hydrophobic contacts. These loops are integral to the ANK fold and do not conform to canonical LIR motifs, thereby defining a unique interface, which differs from the previously reported non-canonical interactions (Huber et al, 2020; von Muhlinen et al, 2012). This binding mode expands our understanding of how ankyrin repeat–containing proteins can selectively engage Atg8 family members, and highlights the structural versatility of GABARAP in recognizing distinct client proteins.

Functionally, ARMS promotes dendritic spine formation and maturation, consistent with its role in scaffolding neurotrophic signals. GABARAP negatively regulates this function: its overexpression suppresses ARMS-induced spine outgrowth, while depletion mimics the ARMS gain-of-function phenotype. Mechanistically, this inhibition requires direct interaction, as ARMS EE mutant fails to rescue spine defects. Domain analyses further show that both GABARAP's binding ability and lipidation status are critical. The L63Q mutant, defective in ARMS binding, has little inhibitory effect, confirming the necessity of direct interaction. In contrast, the lipidation-deficient G116A mutant exerts even stronger inhibition than wild-type GABARAP, suggesting that lipidation enhances degradation of ARMS. Consistently, ARMS WT restores early dendritic branching under ATG7 knockdown but fails to rescue spine loss at later stages, while ARMS EE cannot rescue either phenotype. Together, these findings support a biphasic model in which GABARAP regulates ARMS through both physical sequestration and lipidation-dependent degradation, coordinating its abundance and function across neuronal development.

Furthermore, we show that GABARAP regulates ARMS stability through an autophagy-dependent mechanism. GABARAP overexpression reduces ARMS levels, an effect blocked by chloroquine or by disrupting ARMS-GABARAP binding. Knockdown of either GABARAP or ATG7 leads to ARMS accumulation, indicating that autophagic flux is required for ARMS turnover. This is consistent with the emerging view that scaffold proteins, despite their large size and membrane association, can be selectively degraded via autophagy to ensure optimal protein stoichiometry at synapses. However, ARMS is a membrane-spanning protein localized to intracellular vesicles and the plasma membrane. How GABARAP, classically involved in cytosolic cargo recruitment, mediates the degradation of membrane-associated ARMS remains unclear. It is

possible that ARMS resides in specific endosomal or vesicular compartments that are accessible to GABARAP-mediated autophagic recognition, akin to selective degradation of transmembrane proteins via endophagy. Future studies should explore whether GABARAP targets specific ARMS pools, such as those enriched in growth cones or synaptic boutons, and whether other autophagy receptors assist in this process.

The role of autophagy in regulating scaffolds like ARMS may reflect a general principle of dynamic remodeling at synapses. Tight control of scaffold turnover could enable synapses to respond to activity or developmental cues by adjusting the composition and strength of signaling hubs. Notably, while GABARAP is best known for its autophagic functions, it also participates in GABA_A receptor trafficking, cytoskeletal regulation, and vesicular transport (Chen et al, 2007; Koike et al, 2013). Whether these non-autophagic functions contribute to ARMS regulation remains to be determined. Given the spatial complexity of neurons, it is plausible that GABARAP modulates ARMS behavior in a context-dependent manner.

Finally, our study also explores the potential of disrupting ARMS-GABARAP interactions using ankyrin-derived peptides. The AnkB WT peptide selectively binds GABARAP and outcompetes ARMS for GABARAP binding, leading to the dissociation of the ARMS-GABARAP complex. Notably, expression of AnkB WT in neurons results in increased ARMS accumulation in the soma, suggesting that GABARAP binding regulates ARMS subcellular localization. This highlights the potential of targeting GABARAP-scaffold interactions as a strategy for modulating neuronal protein homeostasis. Moreover, our previous work demonstrates that ankyrin-derived peptides can be used to modulate GABARAP interactions with other synaptic proteins, such as GABA_A receptors (Ye et al, 2021), suggesting broader therapeutic potential for these peptides in regulating synaptic signaling and protein dynamics.

In conclusion, our findings uncover a novel mechanism by which the ARMS-GABARAP interaction regulates dendritic spine development and synaptic maturation. We identify a non-canonical binding mode that underlies GABARAP's selective regulation of ARMS function, providing new insights into scaffold protein regulation in neurons. These results contribute to our understanding of how Atg8 family proteins modulate synaptic plasticity and offer potential therapeutic strategies for diseases linked to dysregulated scaffold protein interactions.

# Methods

### Reagents and tools table

| Reagent/resource | Reference or source | Identifier or catalog number |
| --- | --- | --- |
| **Experimental models** | | |
| HEK-293 cells (*H. sapiens*) | ATCC | CBP60439 |
| Hela (*H. sapiens*) | ATCC | CRM-CCL-2 |
| Hippocampal neurons (*R. rat*) | University of Science and Technology of China | N/A |
| **Recombinant DNA** | | |
| pET32m3c-ARMS 1-13 ARs | This study | N/A |
| pET32m3c-ARMS 1-4 ARs | This study | N/A |
| pET32m3c-ARMS 1-4 ARs mutants | This study | N/A |
| pETm3c-GABARAP | Ye et al (2021) | N/A |
| pETm3c-GABARAP mutants | This study | N/A |
| pETm3c-GABARAPL1 | Ye et al (2021) | N/A |
| pET32m3c-LC3A | This study | N/A |
| pET32m3c-LC3A Q22K, Q26K | This study | N/A |
| pET32m3c-LC3B | This study | N/A |
| pET32m3c-LC3B L22K Q26K | This study | N/A |
| pET32m3c-LC3C | This study | N/A |
| pET32m3c-LC3C G26K | This study | N/A |
| pETm3c-GABARAPL2 | Ye et al (2021) | N/A |
| pCMV-Flag-ARMS WT | This study | N/A |
| pCMV-Flag-ARMS mutants | This study | N/A |
| pCMV-Flag-ARMS 1-13 ARs | This study | N/A |
| pCMV-Flag-ARMS 1-8 ARs | This study | N/A |
| pCMV-Flag-ARMS 1-4 ARs | This study | N/A |
| pCMV-Flag-ARMS 5-13 ARs | This study | N/A |
| pET GST-GABARAP | This study | N/A |
| pET GST | This study | N/A |
| pCMV-GFP-GABARAP | This study | N/A |
| BFP-pll3.7 | This study | N/A |
| BFP-ARMS-shRNA | This study | N/A |
| GFP-GABARAP-shRNA | This study | N/A |
| GFP-ATG7-shRNA | This study | N/A |
| pCMV-GFP-GABARAP L63Q | This study | N/A |
| pCMV-GFP-GABARAP G116A | This study | N/A |

| Reagent/resource | Reference or source | Identifier or catalog number |
| --- | --- | --- |
| pCMV-GFP-LC3B | This study | N/A |
| plko-GABARAP | This study | N/A |
| plko-ATG7 | This study | N/A |
| pCMV-RFP-ARMS | This study | N/A |
| pCMV-RFP-ARMS EE | This study | N/A |
| pET32m3c-AnkB WT | Ye et al (2021) | N/A |
| pET32m3c-AnkB WR | Ye et al (2021) | N/A |
| pCMV-GFP-AnkB WT | Ye et al (2021) | N/A |
| pCMV-GFP-AnkB WR | Ye et al (2021) | N/A |
| **Antibodies** | | |
| Mouse anti-DYKDDDDK | Proteintech | 66008-4-Ig |
| Rabbit anti-GFP | Proteintech | 50430-2-AP |
| Rabbit anti-Kidins220 | Proteintech | 21856-1-AP |
| Mouse anti-RFP | Zenbio | 250120 |
| Rabbit anti-LC3 | Proteintech | 81004-1-PR |
| Mouse anti-P62/SQSTM1 | Proteintech | 18420-1-AP |
| Rabbit-GAPDH | Proteintech | 10494-1-AP |
| Mouse-tubulin | Proteintech | 00068510 |
| Goat anti-rabbit IgG-HRP | Absin | J0609A01 |
| Goat anti-mouse IgG-HRP | Absin | D1230A01 |
| Mouse anti-LAMP1 | SANTA CRUZ | sc-20011 |
| 568 goat anti-rabbit IgG | Invitrogen | 2277758 |
| 488 goat anti-mouse IgG | Invitrogen | 2415945 |
| **Oligonucleotides and other sequence-based reagents** | | |
| PCR primers | This study | Appendix Table S2 |
| **Chemicals, enzymes, and other reagents** | | |
| DMEM F-12 | ThermoFisher | 11320033 |
| Neurobasal | ThermoFisher | 21103049 |
| Glutamax | ThermoFisher | 35050061 |
| B27 | ThermoFisher | 17504044 |
| Trypsin | ThermoFisher | 15090046 |
| HBSS | ThermoFisher | 14175079 |
| Opti-MEM (1×) | ThermoFisher | 31985.70 |
| DMEM basic (1×) | Gibco | C11995500BT |
| Fetal Bovine Serum | Excell Bio | FSP500 |
| CalPhos Mammalian Transfection Kit | Takara | 631312 |
| 4% Paraformaldehyde Fix Solution | Sangon Biotech | E672002 |
| Triton X-100 | Sangon Biotech | A600198 |
| Tween 20 | Sangon Biotech | A600560 |
| NP-40 Lysis Buffer | Beyotime | P0013F |

| Reagent/resource | Reference or source | Identifier or catalog number |
|---|---|---|
| RIPA Lysis Buffer | Beyotime | P0013B |
| 2×Phanta Master Mix | Vazyme | P525-01 |
| ClonExpress II One Step Cloning Kit | Vazyme | C112-02 |
| 10×DNA loading buffer | Vazyme | P022-01 |
| DNA marker | Vazyme | MD102-02 |
| BamHI-HF | NEB | R3136L |
| EcoRI-HF | NEB | R3101L |
| XhoI | NEB | R0146L |
| AgeI-HF | NEB | R3552S |
| CutSmart | NEB | B7204S |
| Proteinase K | MERCK | 03115879001 |
| Chloroquine | MCE | HY-17589A |
| Trichloroacetic acid | Aladdin | T104257 |
| Acetone | Sinopharm Chemical Reagent Company | 10000418 |
| HCl | Sinopharm Chemical Reagent Company | 10011028 |
| NON-FAT Powdered Milk | Sangon Biotech | A600669-0250 |
| IPTG | Sangon Biotech | A600168-0100 |
| DTT | Sangon Biotech | A620058-0100 |
| **Software** | | |
| GraphPad Prism | https://www.graphpad.com/ | |
| ImageJ | https://imagej.net/ij/index.html | |
| IMARIS 9.6.2 | https://imaris.oxinst.cn/versions/9-6 | |
| **Other** | | |
| ZEISS LSM980 with Airyscan | ZEISS | |
| VP-ITC | Malvern | |
| Olympus SZX16 | Olympus | |
| Tanon 5200 Multi | Tanon | |

## Constructs, protein expression and purification

The coding sequences of the rat ARMS (UniProt: Q9EQG6) constructs were PCR amplified from cDNA libraries. Various mutations or shorter fragments of ARMS and GABARAP were generated using standard PCR-based methods and confirmed by DNA sequencing. Sequences of ARMS, GABARAP subfamily proteins, and LC3 subfamily proteins were cloned into a modified pET32a vector containing a thioredoxin (Trx)-His$_6$ tag, or a His$_6$ tag or a pGEX4T-1 vector containing a GST tag at the N-terminus for protein expression. The primer sequences used in this study are listed in Appendix Table S2. The N-terminal Trx-His$_6$-tagged proteins, N-terminal His$_6$-tagged proteins, and GST-tagged proteins were expressed in *Escherichia coli* BL21 (DE3) cells in LB medium, induced by IPTG at 16 °C for 20 h. Proteins were then purified using Ni-NTA agarose for His$_6$-tagged proteins and GSH-Sepharose affinity chromatography for GST-tagged proteins, followed by size exclusion chromatography (Superdex 200 or Superdex 75) in a buffer containing 100 mM NaCl, 50 mM Tris, 1 mM EDTA, and 1 mM DTT, pH 7.8. Trx-His$_6$-tags were removed by incubation with human rhinovirus 3 C protease at 4 °C overnight, and the protease was subsequently separated by size exclusion chromatography.

## FPLC assay

The analysis was performed using an AKTA FPLC system (GE Healthcare). Proteins with N-terminal Trx-His$_6$ tags and N-terminal His$_6$-tags were concentrated to 100 μM and loaded onto a Superdex 200 increase column (GE Healthcare), equilibrated with buffer containing 100 mM NaCl, 50 mM Tris, 1 mM EDTA, and 1 mM DTT, pH 7.8. Data were analyzed using Origin 7.0.

## Isothermal titration calorimetry (ITC) assay

ITC measurements were carried out on a VP-ITC Microcal calorimeter (Malvern) at 25 °C. Proteins were dissolved in a buffer containing 100 mM NaCl, 50 mM Tris, 1 mM EDTA, and 1 mM DTT, pH 7.8. Titrations were conducted by injecting 10 μL aliquots of syringe protein into the cell at 180-second intervals to allow the peak to return to baseline. Atg8 family proteins and ankyrin peptides (200 μM or 300 μM) were loaded into the syringe, and various fragments of ARMS proteins (20 μM) were loaded into the cell. The titration data were fitted using the one-site binding model and analyzed with Origin 7.0.

## GST pull-down assay

Freshly purified GST-GABARAP protein and Trx tagged ankyrin peptides were used in the assay. Flag-tagged WT and various fragments of Flag-ARMS ARs were overexpressed in HEK293T cells. Cells were harvested and lysed in cold NP40 lysis buffer (P0013F, Beyotime), followed by centrifugation at 14,000 rpm for 10 min at 4 °C. The supernatants were incubated with 2 μM purified proteins and the same volume of supernatant for 1 h at 4 °C. Then, 50 μL GSH-Sepharose 4B slurry beads were added and incubated for 1 h at 4 °C. After washing three times with PBS buffer, the captured proteins were eluted by boiling with 20 μL SDS-PAGE loading dye and analyzed by western blot.

## Immunoblotting

Cells were lysed using NP40 lysis buffer (P0013F, Beyotime) and RIPA lysis buffer (P0013B, Beyotime), and the lysates were incubated with anti-Flag beads (Sigma, A2220) for 2 h at 4 °C. After washing three times with lysis buffer, the beads were incubated in SDS-PAGE loading buffer for 15 min. The samples were separated on SDS-PAGE and transferred onto a PVDF membrane (Millipore, Sigma-Aldrich), probed with GFP (1:1000; Proteintech, 50430-2-AP) or Flag (1:2000; Proteintech, 66008-4-Ig) antibodies at 4 °C overnight, followed by incubation with secondary antibodies at room temperature for 1 h.

## Immunoblotting statistics

For both the pull-down and immunoblotting analyses, we performed at least three independent experiments. Data were statistically analyzed using one-way ANOVA with Dunnett's test to compare test samples to the control, and results are presented as mean ± SEM. The threshold for statistical significance was set at $P < 0.05$ for all tests. Data analysis was conducted using ImageJ and GraphPad Prism version 9.5.0. Source data are included with this paper.

## Circular dichroism measurements

Trx-ARMS 1-4 ARs and mutant proteins (E33R, E67R, and E33R E67R) were purified and diluted into 0.2 mg/mL in a buffer containing 10 mM NaCl. The Circular dichroism measurements (CD) spectra of the proteins were acquired on a JASCO J-1700 CD Spectrometer at the room temperature, and the bandwidth is set to 2 nm. Each spectrum was collected with three scans spanning a spectral window of 190–260 nm.

## Crystallography and data analysis

The purified ARMS ARs-GABARAP fusion protein were held together by a linker, which consists of eight amino acids: GSLVPRGS. Crystals of the ARMS ARs-GABARAP fusion protein were obtained using the hanging drop diffusion method at 16 °C in a buffer containing 45% (w/v) PEG, 100 mM MES, and 400 mM potassium chloride. Diffraction data were collected at the Shanghai Synchrotron Radiation Facility and processed using HKL2000. Structures were solved by molecular replacement using GABARAP (PDB: 1KJT) as the model. Final refinement statistics are summarized in Appendix Table S1. All structure figures were prepared using PyMOL software (http://www.pymol.org). The coordinates and structure factors of the ARMS-GABARAP complex have been deposited in the Protein Data Bank with accession number 9L9I.

## Triton solubility assay

293T cells or neurons were lysed on ice in lysis buffer containing 1% Triton X-100, 50 mM Tris-HCl (pH 7.8), and 150 mM NaCl for 30 min to ensure complete lysis. Lysates were then centrifuged at $12,000\times g$ for 10 min at 4 °C to separate soluble and insoluble fractions. The supernatant (soluble fraction) was carefully transferred to a new tube. The remaining pellet (insoluble fraction) was washed twice with 500 µL of ice-cold lysis buffer to remove residual soluble proteins, followed by centrifugation at $12,000\times g$ for 10 min at 4 °C. Equal volumes of 2× SDS-PAGE loading buffer were added to both the supernatant and the washed pellet, and samples were boiled at 80 °C for 10 min. Proteins from soluble and insoluble fractions were separated by SDS-PAGE, analyzed by immunoblotting described in the "Immunoblotting" section

## Protease protection assay

HEK293T cells transfected with Flag-ARMS plasmid were treated with 30 µM CQ for 10 h at 24 h post-transfection. Cells were lysed on ice for 10 min in buffer containing 250 mM sucrose, 10 mM HEPES (pH 7.4), 0.1% Triton X-100. Crude autophagosomes were isolated by differential centrifugation ($1000\times g$ for 10 min first, then $13,000\times g$ for 20 min) and washed twice with sucrose-HEPES buffer. The resuspended pellet was split into four aliquots: (1) protease inhibitors; (2) protease inhibitors; (2) proteinase K (80 µg/mL); (3) proteinase K and 0.4% Triton X-100. After 30 min on ice, reactions were terminated with 10% TCA, centrifuged ($13,000\times g$ for 10 min), and pellets were washed twice with acetone. Pellets resuspended in SDS loading buffer: one protease inhibitor sample was loaded unboiled, others boiled at 80 °C for 10 min, followed by immunoblotting as described.

## HEK293T cells culture

HEK293T cells were cultured in 35 mm disposable Corning dishes in DMEM supplemented with 10% fetal bovine serum and penicillin-streptomycin (100 U/mL) at 37 °C in a humidified atmosphere with 5% $CO_2$. Transfection of HEK293T cells was performed using PEI when the cell density reached 80%. The medium was changed every 24 h, and cells were fixed for Co-IP or immunostaining 36 h after transfection.

## Hippocampal neuronal culture and transfection

All animal work was conducted in accordance with the guidelines of the University of Science and Technology of China Animal Care and Use Committee, with approval number USTCACUC1801002. Primary hippocampal neurons were isolated from E18 SD rat embryos. Hippocampi were dissected and digested with 0.25% trypsin (Life Technologies), then plated onto Poly-L-Lysine-coated glass coverslips in culture dishes. Neurons were incubated at 37 °C in Neurobasal medium (Life Technologies) supplemented with 2% B27, 0.5 mM glutamine, 12.5 µM glutamate, and 1% penicillin. Transfection of hippocampal neurons was performed on day 4 or day 10 using calcium phosphate-DNA coprecipitation, followed by fixation and immunostaining on days 7, 14, or 18. ARMS-shRNA was cloned into the BFP-pll3.7 vector with a sequence targeting ARMS (5'-gccaccaagatgagaaata-3'). ARMS rescue plasmids were cloned into a RFP-tagged vector. Hippocampal neurons were transfected with ARMS-shRNA or rescue plasmids on day 10, and fixed on days 14 or 18.

## Antibodies and confocal imaging

Rabbit polyclonal antibodies against Flag tag (1:1000; Proteintech, 20543-1-AP), Kidins220/ARMS (1:500; Proteintech, 21856-1-AP), and GOLGA2/GM130 (1:1000; Proteintech, 66662-1-Ig) were used for immunofluorescence. Secondary antibodies conjugated to Alexa Fluor 568 (1:1000; ThermoFisher, 2500544) or Alexa Fluor 647 (1:1000; ThermoFisher, 2555716) were used for detection. Neurons were fixed for 15 min at room temperature with 4% paraformaldehyde, then treated with 0.2% Triton X-100 and blocked with 2% BSA. Primary antibodies were diluted in blocking buffer (2% BSA, 0.2% Triton X-100, 0.1% Tween 20) and incubated overnight at 4 °C. Secondary antibodies were applied at room temperature for 1 h, followed by PBST washes. After mounting with DAPI, the slides were examined under a Zeiss LSM980 laser-scanning confocal microscope using ×40 or ×63 oil immersion objectives. Fluorescence image processing was performed using ImageJ

software. Z stacking was performed with 0.5 μm steps in the z direction, and 512 × 512-pixel resolution images were reconstructed using IMARIS 9.6.2 software (https://imaris.oxinst.cn/versions/9-6). The soma intensity ratios in neurons were quantified used IMARIS, the data were analyzed using GraphPad Prism.

### Dendritic spine quantification

Dendritic spine density and maturation were quantified from confocal microscopy images using ImageJ. Total dendritic length and spine number were measured, and spine density was calculated as the number of spines per 10 μm of dendrite. Spine maturity was assessed by classifying spines as mushroom-shaped and quantifying the proportion of mature spines. We performed at least three independent experiments, data analysis was conducted using ImageJ and GraphPad Prism. Data were statistically analyzed using one-way ANOVA with Dunnett's test to compare test samples to the control, and results are presented as mean ± SEM. The threshold for statistical significance was set at $P < 0.05$ for all tests.

## Data availability

All data needed to evaluate the conclusions in the paper are present in the paper, and/or the Expanded View (EV) files. The atomic coordinates and structure factor data of the ARMS-GABARAP complex have been deposited to the Protein Data Bank under the accession code 9L9I.

The source data of this paper are collected in the following database record: biostudies:S-SCDT-10_1038-S44318-025-00669-w.

## Peer review information

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

## Acknowledgements

We thank beamlines BL18U1, BL19U1, and BL02U1 at Shanghai Synchrotron Radiation Facility (SSRF, China) for X-ray beam time. This work was supported by grants from the National Natural Science Foundation of China (32521003, 32100764, 32470808, 22122703, 32170767, and 32541067), the Major Frontier Research Project (LS9100000002) and the Double First-Class Initiative (YD9100002507) of the University of Science and Technology of China, the Fundamental Research Funds for the Central Universities (WK2490250006), the Center for Advanced Interdisciplinary Science and Biomedicine of IHM (QYPY20220014), and the Strategic Priority Research Program of the Chinese Academy of Sciences (XDB0490000).

## Author contributions

**Wenli Jiang**: Conceptualization; Data curation; Formal analysis; Validation; Investigation; Visualization; Methodology; Writing—original draft; Writing—review and editing. **Jin Ye**: Data curation; Formal analysis; Funding acquisition; Validation; Investigation. **Jiasheng Chen**: Investigation; Structure determination. **Xinyu Wang**: Data curation; Formal analysis; Investigation; Methodology. **Yahong Li**: Formal analysis; Validation; Investigation. **Jianchao Li**: Formal analysis; Validation; Structure determination. **Yide Mei**: Resources; Formal analysis; Validation. **Yanlu Lyu**: Resources; Formal analysis; Validation. **Wei Hu**: Resources; Formal analysis; Validation. **Chao Wang**: Conceptualization; Resources; Formal analysis; Supervision; Funding acquisition; Validation; Writing—original draft; Writing—review and editing.

Source data underlying figure panels in this paper may have individual authorship assigned. Where available, figure panel/source data authorship is listed in the following database record: biostudies:S-SCDT-10_1038-S44318-025-00669-w.

## Disclosure and competing interests statement

The authors declare no competing interests.

# Expanded View Figures

**Figure EV1.   ARMS 1-4 ARs specifically interact with the GABARAP subfamily but not LC3s.**

(**A**) Multiple sequence alignment was performed using ARMS/KIDINS220 proteins from Homo sapiens (human), Rattus norvegicus (rat), Mus musculus (mouse), Cavia porcellus (guinea pig), and Danio rerio (zebrafish). Absolutely conserved and highly conserved residues are highlighted in dark red and light red, respectively. The predicted LIR-like motifs are marked with green boxes, while the blue stars indicate critical residues identified in this study as essential for binding to GABARAP. The 13 ankyrin repeats (ARs 1-13) are manually defined and labeled. This alignment highlights the strong conservation of the N-terminal cytoplasmic region across species, particularly in the ankyrin repeat domain and GABARAP-interacting interface. (**B**) FPLC assay showing that ARMS 1-13 ARs interacts with GABARAP. (**C, D**) FPLC assay showing that ARMS 1-4 ARs interact with GABARAPL1 (**C**) and GABARAPL2 (**D**). (**E**) Co-immunoprecipitation analysis of Flag-tagged ARMS with Atg8 family proteins in HEK293T cells. ARMS binds to GABARAP and GABARAPL1, but not to GFP-LC3A or GFP-LC3B, confirming GABARAP subfamily specificity. (**F**) Summary table of ITC-derived binding affinities between ARMS and various Atg8 family proteins. (**G–I**) FPLC assay showing that ARMS 1-4 ARs do not interact with LC3A (**G**), LC3B (**H**), or LC3C (**I**), further supporting subfamily-specific binding. Source data are available online for this figure.

▶

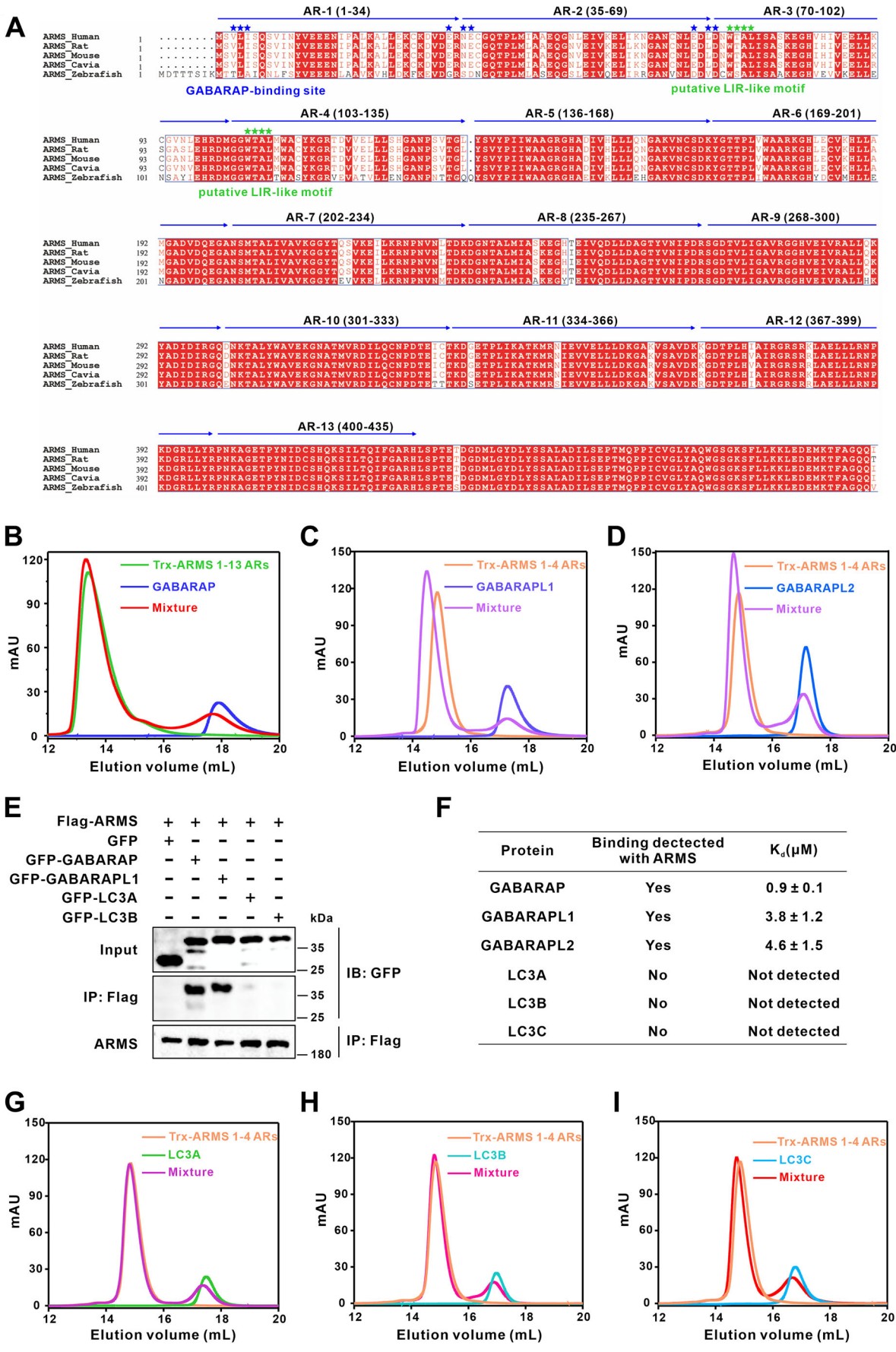

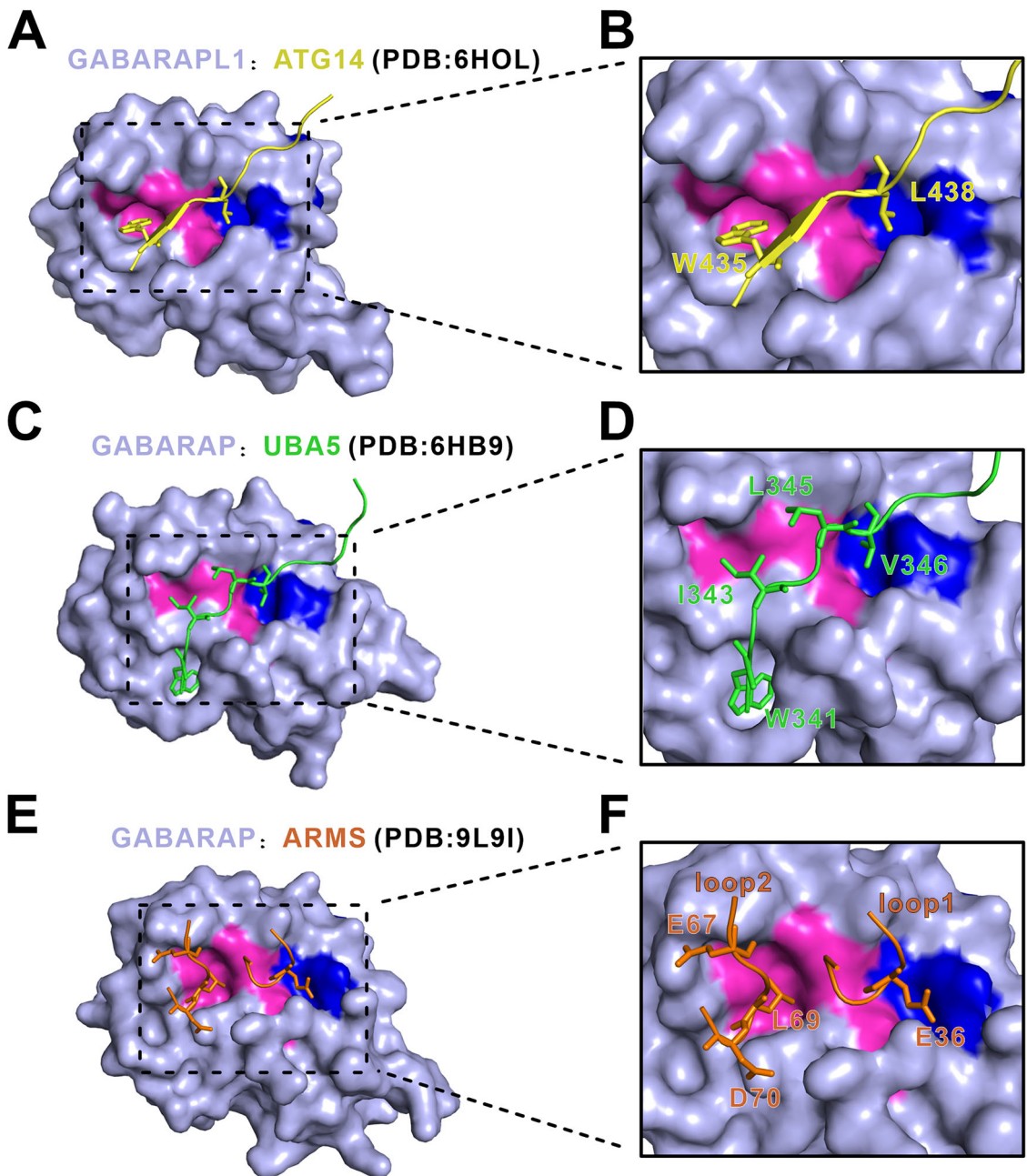

**Figure EV2.  ARMS 1-4 ARs interact with GABARAP via a non-canonical binding mode.**

(A, B) Crystal structure of GABARAPL1 in complex with the ATG14 LIR peptide (PDB: 6HOL). GABARAPL1 is shown in light blue, and ATG14 LIR peptide is shown in yellow. ATG14 W435 is deeply inserted into HP1 (pink), while L438 occupies HP2 (blue). (C, D) Crystal structure of GABARAP in complex with the UBA5 LIR peptide (PDB: 6HB9). GABARAP is shown in light blue, and the UBA5 LIR peptide in green. UBA5 I343, L345, and V346 interact with HP1 (pink) and HP2 (blue), while W341 binds near α-helix α1 of GABARAP (HP0). (E, F) Crystal structure of GABARAP in complex with ARMS (PDB: 9L9I). GABARAP is shown in light blue, while loop1 and loop2 of ARMS (orange) insert into HP1 (pink) and HP2 (blue), suggesting a distinct interaction mode compared to canonical LIR binding.

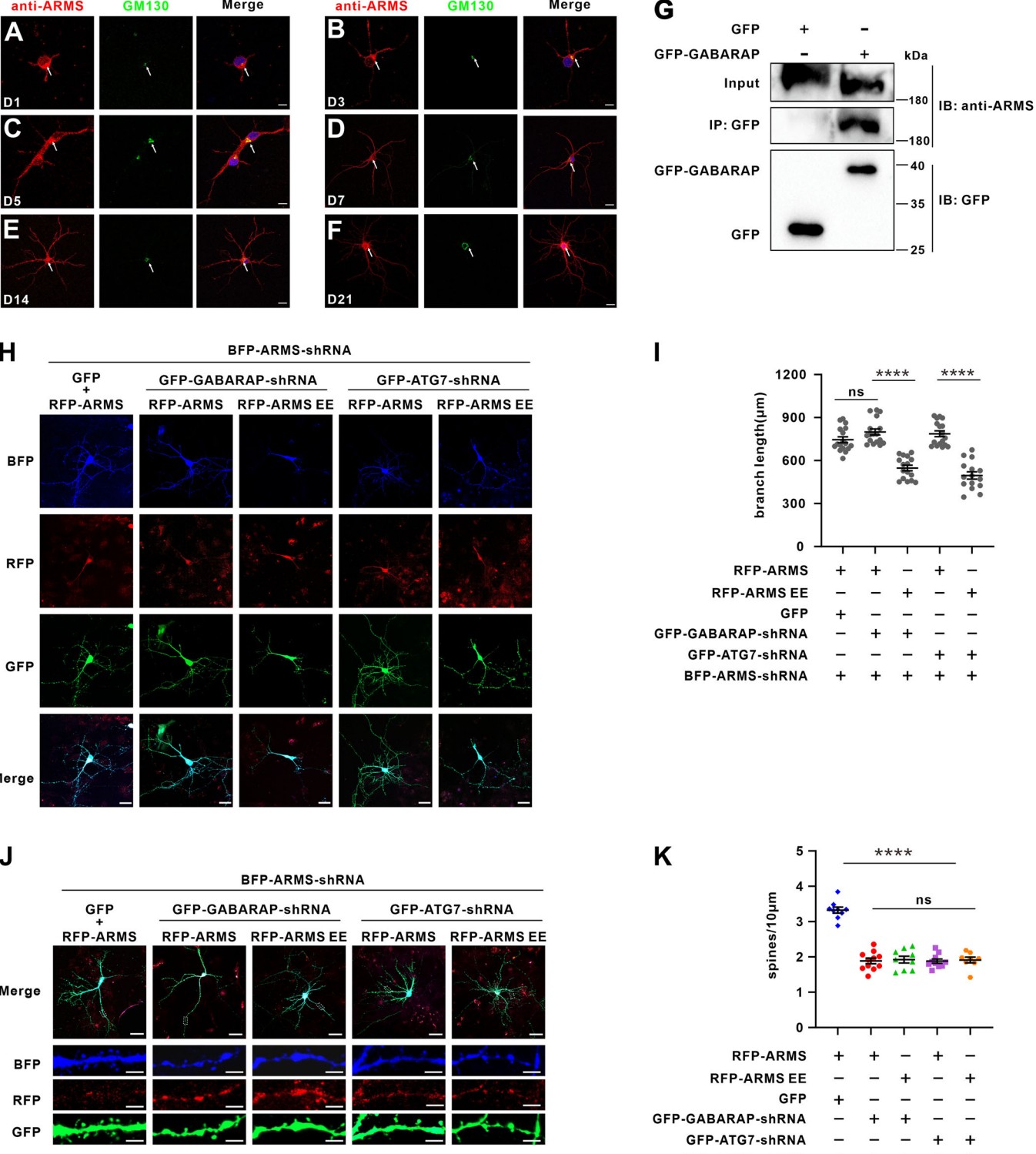

◀ **Figure EV3. GABARAP and ATG7 regulate the physiological functions of ARMS in neuron.**

(A–F) Confocal images showing the developmental localization of endogenous ARMS in cultured hippocampal neurons. Neurons were fixed at day 1 (A), day 3 (B), day 5 (C), day 7 (D), day 14 (E), and day 21 (F), and stained for ARMS (red) and the Golgi marker GM130 (green). ARMS is initially localized in the soma and progressively redistributed to dendrites and axons as neurons mature, with prominent accumulation near the Golgi. Scale bars: (A–C) 10 μm; (D–F) 20 μm. (G) Co-immunoprecipitation of endogenous ARMS with GFP-GABARAP in primary hippocampal neurons (DIV7). Neurons were transfected at day 5 with GFP or GFP-GABARAP, and lysates were subjected to GFP pull-down at day 7. Western blotting confirmed interaction between ARMS and GABARAP. (H) Representative confocal images of hippocampal neurons transfected at day 4 with BFP-ARMS-shRNA, and co-transfected with one of the following: GFP and RFP-ARMS WT; GFP-GABARAP-shRNA and RFP-ARMS WT or RFP-ARMS EE; GFP-ATG7-shRNA and RFP-ARMS WT or RFP-ARMS EE. Neurons were fixed and imaged at day 7. ARMS WT fully rescued dendritic arborization even under GABARAP or ATG7 knockdown, whereas the EE mutant failed to do so. Scale bar, 30 μm. (I) Quantification of neurite length under each condition. Sample sizes: GFP-pll3.7 and RFP-ARMS, $n = 17$; GFP-GABARAP-shRNA and RFP-ARMS, $n = 17$; GFP-GABARAP-shRNA and RFP-ARMS EE, $n = 16$; GFP-ATG7-shRNA and RFP-ARMS, $n = 17$; GFP-ATG7-shRNA and RFP-ARMS EE, $n = 15$. Error bars indicate SEM. Data were normally distributed, and one-way ANOVA was used for statistical analysis. ****$P < 0.0001$, ns, not significant. (J) Representative images of hippocampal neurons transfected at day 10 with BFP-ARMS-shRNA, and co-transfected with the same five experimental conditions as in (A). Neurons were fixed and imaged at day 14. ARMS WT failed to rescue dendritic spine density under GABARAP or ATG7 knockdown, and ARMS EE showed no rescue in all conditions. Upper panels: low magnification (scale bar, 50 μm); lower panels: high magnification of dendritic spines (scale bar, 5 μm). (K) Quantification of spine density from three independent experiments. Sample sizes: GFP and RFP-ARMS, $n = 9$; GABARAP-shRNA and RFP-ARMS, $n = 11$; GABARAP-shRNA and RFP-ARMS EE, $n = 9$; ATG7-shRNA and RFP-ARMS, $n = 10$; ATG7-shRNA and RFP-ARMS EE, $n = 8$. Error bars indicate SEM. Data were normally distributed, and one-way ANOVA was used for statistical analysis. ****$P < 0.0001$, ns, not significant. Source data are available online for this figure.

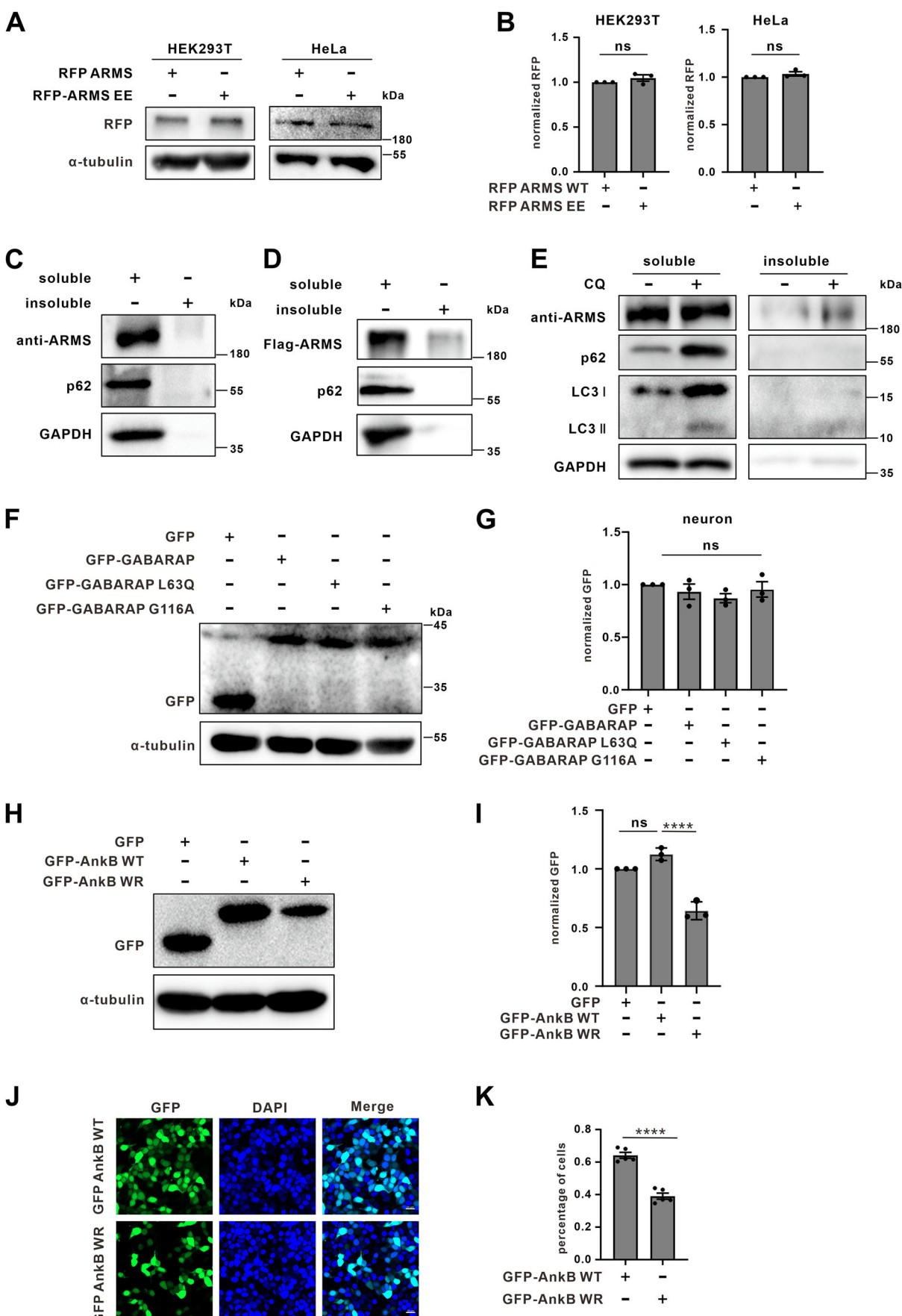

**Figure EV4. Plasmid-expressed proteins exhibit normal expression and solubility in cells.**

(A) Western blot analysis of HEK293T and HeLa cells overexpressing RFP-ARMS WT or RFP-ARMS EE, showing comparable expression levels. (B) Quantification of ARMS expression from three independent experiments ($n = 3$). Data are presented as mean ± SEM. Statistical significance: ns, not significant. Data were analyzed by unpaired two-tailed $t$ test. (C) Lysed HEK293T cells using lysis buffer containing 1% Triton X-100, followed by western blot analysis of the resulting supernatant and pellet fractions. The results demonstrated that endogenous ARMS protein was almost exclusively detected in the supernatant. (D) Overexpressed Flag-ARMS in HEK293T cells, lysed cells using lysis buffer containing 1% Triton X-100, followed by western blot analysis of the resulting supernatant and pellet fractions. The results demonstrated that overexpressed ARMS protein was predominantly detected in the supernatant, with only a faint band observed in the insoluble fraction. (E) Neurons treated with chloroquine (CQ) for 12 h were lysed in 1% Triton X-100 buffer. Western blot of soluble and insoluble fractions revealed that ARMS, p62, LC3-I/II, and GAPDH serving as a soluble loading control were almost exclusively detected in the soluble fraction, indicating the absence of aggregation. (F, G) GFP, GFP-GABARAP WT and mutants (L63Q or G116A) were transfected into primary hippocampal neurons. Western blot analysis (F) and quantification (G) showed comparable expression levels across all constructs ($n = 3$). Data represent mean ± SEM. Data were normally distributed and analyzed by one-way ANOVA. ns: not significant. (H, I) GFP, GFP-AnkB WT or WR were transfected into neurons. Western blot analysis revealed significantly reduced expression of the WR construct (H). Quantification confirmed the difference in protein levels (I). Data represent mean ± SEM. Data were normally distributed and analyzed by one-way ANOVA. ****$P < 0.0001$, ns: not significant. (J) To assess whether reduced WR expression was due to lower transfection efficiency, GFP-AnkB WT and WR were transfected into HEK293T cells. Fluorescence-based quantification of GFP-positive cells showed significantly lower transfection efficiency for WR (J), explaining the reduced protein levels observed in neurons. Data represent mean ± SEM; ****$P < 0.0001$. (K) Quantification of transfected cell ratios from five independent replicates ($n = 5$) of (J). Data are presented as mean ± SEM. Data were analyzed by unpaired two-tailed $t$ test. ****$P < 0.0001$. Source data are available online for this figure.

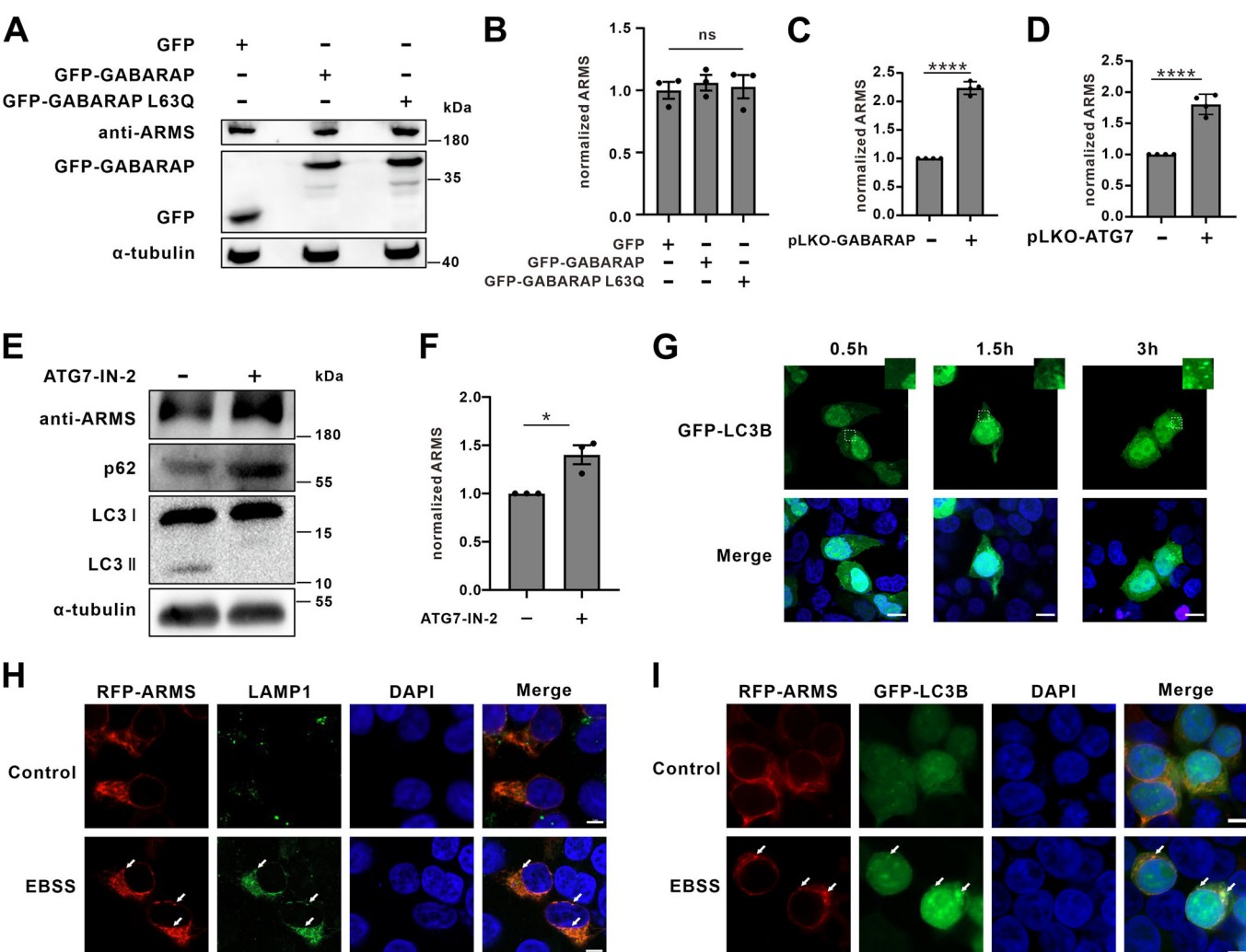

**Figure EV5. Autophagy affects the expression and localization of ARMS in cells.**

(A) Western blot analysis of HEK293T cells overexpressing GFP, GFP-GABARAP WT, or GFP-GABARAP L63Q shows that neither GABARAP nor its mutant affects endogenous ARMS protein levels. (B) Quantification of ARMS levels from three independent experiments ($n = 3$). Data are presented as mean ± SEM. Data were normally distributed and analyzed by one-way ANOVA. ns, not significant. (C) Endogenous GABARAP was knocked down in HEK293T cells using shRNA. Quantification of ARMS levels from four independent experiments ($n = 4$) shows a significant increase. Data are presented as mean ± SEM. Data were analyzed by unpaired two-tailed $t$ test. ****$P < 0.0001$. (D) Endogenous ATG7 was knocked down in HEK293T cells using shRNA. Quantification of ARMS levels from four independent experiments ($n = 4$) also shows a significant increase. Data are presented as mean ± SEM. Data were analyzed by unpaired two-tailed $t$ test. ****$P < 0.0001$. (E) Neurons were treated with the ATG7 inhibitor ATG-IN-2 for 24 h. Western blot analysis revealed elevated ARMS levels. (F) Quantification of ARMS levels from three independent experiments ($n = 3$). Data are presented as mean ± SEM. Data were analyzed by unpaired two-tailed $t$ test. *$P = 0.0148$. (G) GFP-LC3B was expressed in HEK293T cells and autophagy was induced by EBSS starvation for 0.5, 1, or 3 h. Immunofluorescence analysis revealed time-dependent accumulation of LC3B puncta after 3 h (scale bar, 10 μm). White boxes indicate the regions enlarged in the upper-right corner. (H) HEK293T cells expressing RFP-ARMS were subjected to EBSS treatment for 3 h, followed by immunostaining with an antibody against LAMP1 (lysosomal marker), revealing partial co-localization (scale bar, 5 μm). White arrows indicate RFP-ARMS and LAMP1 co-localized puncta. (I) Co-expression of RFP-ARMS and GFP-LC3B in HEK293T cells followed by EBSS treatment for 3 h showed that ARMS puncta partially co-localized with LC3B puncta (scale bar, 5 μm). White arrows indicate RFP-ARMS and GFP-LC3B co-localized puncta.

