## [Peer Review File · The EMBO Journal]

A non-canonical ARMS-GABARAP interaction modulates dendritic spine formation and synaptic development

Wenli Jiang, Jin Ye, Jiasheng Chen, Xinyu Wang, Yahong Li, Jianchao LI, Yide Mei, Yanlu Lyu, Wei Hu, and Chao WANG

Corresponding author(s): Chao WANG (cwangust@ustc.edu.cn) , Yanlu Lyu (drlv_ent@outlook.com), Wei Hu (andinghu@ustc.edu.cn), Jin Ye (jinye123@ustc.edu.cn)

Review Timeline:

Submission Date:	5th May 25
Editorial Decision:	24th Jun 25
Revision Received:	4th Nov 25
Editorial Decision:	19th Nov 25
Revision Received:	19th Nov 25
Accepted:	24th Nov 25

Editor: William Teale

Transaction Report:

Dear Dr. Wang,

Thank you again for the submission of your manuscript entitled "A noncanonical ARMS-GABARAP interaction modulates dendritic spine formation and synaptic development" and for your patience during the review process. We have now received the reports from three referees, which I copy below.

As you can see from their comments, all referees thought your manuscript was interesting and timely. While referees #2 and #3 raised concerns over the physiological context and functional relevance of your data, the points they raise seem reasonable and addressable.

Therefore, based on the overall interest expressed in the reports, I would like to invite you to address the comments of all referees in a revised version of the manuscript. I should add that it is The EMBO Journal policy to allow only a single major round of revision and that it is therefore important to resolve the main concerns at this stage. Please contact me if you have any questions, need further input on the referee comments or if you anticipate any problems in addressing any of their points; I am available at any time to discuss your revision plan over Zoom. Please, follow the instructions below when preparing your manuscript for resubmission.

I would also like to point out that as a matter of policy, competing manuscripts published during this period will not be taken into consideration in our assessment of the novelty presented by your study ("scooping" protection). We have extended this 'scooping protection policy' beyond the usual 3 month revision timeline to cover the period required for a full revision to address the essential experimental issues. Please contact me if you see a paper with related content published elsewhere to discuss the appropriate course of action.

Again, please contact me at any time during revision if you need any help or have further questions.

Thank you very much again for the opportunity to consider your work for publication. I look forward to your revision.

Best regards,

William

William Teale, Ph.D.
Editor
The EMBO Journal

When submitting your revised manuscript, please carefully review the instructions below and include the following items:

- 1) a .docx formatted version of the manuscript text (including legends for main figures, EV figures and tables). Please make sure that the changes are highlighted to be clearly visible.
- 2) individual production quality figure files as .eps, .tif, .jpg (one file per figure).
- 3) a .docx formatted letter INCLUDING the reviewers' reports and your detailed point-by-point response to their comments. As part of the EMBO Press transparent editorial process, the point-by-point response is part of the Review Process File (RPF), which will be published alongside your paper.
- 4) a complete author checklist, which you can download from our author guidelines ([https://wol-prod-cdn.literatumonline.com/pb-assets/embo-site/Author Checklist%20-%20EMBO%20J-1561436015657.xlsx](https://wol-prod-cdn.literatumonline.com/pb-assets/embo-site/Author%20Checklist%20-%20EMBO%20J-1561436015657.xlsx)). Please insert information in the checklist that is also reflected in the manuscript. The completed author checklist will also be part of the RPF.
- 5) Please note that all corresponding authors are required to supply an ORCID ID for their name upon submission of a revised manuscript.
- 6) We require a 'Data Availability' section after the Materials and Methods. Before submitting your revision, primary datasets produced in this study need to be deposited in an appropriate public database, and the accession numbers and database listed under 'Data Availability'. Please remember to provide a reviewer password if the datasets are not yet public (see

<https://www.embopress.org/page/journal/14602075/authorguide#datadeposition>). If no data deposition in external databases is needed for this paper, please then state in this section: This study includes no data deposited in external repositories. Note that the Data Availability Section is restricted to new primary data that are part of this study.

Note - All links should resolve to a page where the data can be accessed.

8) For data quantification: please specify the name of the statistical test used to generate error bars and P values, the number (n) of independent experiments (specify technical or biological replicates) underlying each data point and the test used to calculate p-values in each figure legend. The figure legends should contain a basic description of n, P and the test applied. Graphs must include a description of the bars and the error bars (s.d., s.e.m.).

9) We would also encourage you to include the source data for figure panels that show essential data. Numerical data can be provided as individual .xls or .csv files (including a tab describing the data). For 'blots' or microscopy, uncropped images should be submitted (using a zip archive or a single pdf per main figure if multiple images need to be supplied for one panel). Additional information on source data and instruction on how to label the files are available at .

10) We replaced Supplementary Information with Expanded View (EV) Figures and Tables that are collapsible/expandable online (see examples in <https://www.embopress.org/doi/10.15252/embj.201695874>). A maximum of 5 EV Figures can be typeset. EV Figures should be cited as 'Figure EV1, Figure EV2" etc. in the text and their respective legends should be included in the main text after the legends of regular figures.

12) Our journal encourages inclusion of *data citations in the reference list* to directly cite datasets that were re-used and obtained from public databases. Data citations in the article text are distinct from normal bibliographical citations and should directly link to the database records from which the data can be accessed. In the main text, data citations are formatted as follows: "Data ref: Smith et al, 2001" or "Data ref: NCBI Sequence Read Archive PRJNA342805, 2017". In the Reference list, data citations must be labeled with "[DATASET]". A data reference must provide the database name, accession number/identifiers and a resolvable link to the landing page from which the data can be accessed at the end of the reference. Further instructions are available at .

13) In order to increase the reproducibility and reach of your work, The EMBO Journal includes a table of reagents that were used in the study. Please provide this along with your revisions.

Further instructions for preparing your revised manuscript:

We realize that it is difficult to revise to a specific deadline. In the interest of protecting the conceptual advance provided by the work, we recommend a revision within 3 months (22nd Sep 2025). Please discuss the revision progress ahead of this time with the editor if you require more time to complete the revisions. Use the link below to submit your revision:

Referee #1:

Jiang et al report a detailed biochemical and structural characterization of the interaction between the scaffolding protein ARMS and GABARAP. They show by solving the crystal structure of a fusion protein of the first 4 ankyrin repeats and GABARAP a new interaction mode that does not rely on any classical LIR peptides. Instead, the two classical hydrophobic pockets of GABARAP are recognized by a network of polar and hydrophobic interactions. The authors validate these by mutagenesis and show that interaction with the LC3 subfamily of ATG8 proteins does not occur unless LC3s get mutated to allow critical interactions that are present in GABARAP. The authors also show that these interactions are functionally important in cellular assays. Overall, this is an interesting and carefully designed study. My only comment is that it is not quite clear to me how autophagy regulates the protein level. ARMS is a membrane protein and including more information in which membranes it occurs and if autophagy includes removal part of the membranes as well or if a protein retrieval mechanism into the cytoplasm is required. Adding this to the discussion would strengthen the manuscript.

Referee #2:

Summary

In this manuscript, Jiang et al. examined how the neuronal scaffold protein ankyrin repeat-rich membrane spanning (ARMS) is regulated by its binding partners. Firstly, the authors found that members of the GABARAP subfamily of ATG8 proteins directly bind to ARMS with low μM affinities. Binding site mapping revealed that this interaction is mediated by the first four ankyrin repeats of ARMS (1-4 ARMS ARs). Surprisingly, the crystal structure of ARMS binding region in complex with GABARAP showed that the first two finger-like loops connecting adjacent ankyrin repeats insert into GABARAP's hydrophobic pockets (HP1, HP2) via prominent electrostatic interactions. Intriguingly, key residues involved in these interactions are absent in LC3 members of the ATG8 family, explaining the lack of interaction with LC3 proteins. Next, the authors sought to assess the significance of the ARMS-GABARAP binding in neuronal cells and found that rescue of ARMS knockdown (KD) with a GABARAP-binding deficient variant (EE) fails to restore dendritic growth and spine density to the level of wild-type ARMS. Lastly, the authors showed that GABARAP-binding to ARMS controls its abundance. Collectively, the work of Jiang and colleagues clearly establish ARMS as new ATG8 interacting protein and provide evidence for an exciting new LIR independent, non-canonical binding mode. However, the functional relevance of this interaction is less clear. Several critical points remain to be addressed to strengthen this part of the study.

- 1) Does ARMS interact with GABARAP in neurons? It is not evident how 293T cells help to assess the ARMS-GABARAP interaction in a physiological context (Figure 4A).
- 2) Does KD of GABARAP phenocopy the effect of ARMS EE re-expression in ARMS KD with regard to dendritic growth and spine density (Figure 4B and 4D)? In other words, what happens to dendrites and spines when GABARAP is depleted during ARMS WT re-expression? Along similar lines, the authors should assess the effect of ATG7 inhibition. Does this prevent the effect of ARMS WT re-expression?
- 3) The authors need to show by Western blot that ARMS WT and EE are expressed to similar levels in these neurons (Figure 4B and 4D). The same applies to Figure 5 and GFP and GFP-GABARAP as well as to Figure 7F and GFP-AnkB WT and WR.
- 4) What is the effect of ARMS-binding deficient GABARAP variant (e.g. L63Q) expression on spine density and maturation? The authors should expand Figure 5E and 5F by including GABARAP L63Q and compare it to GABARAP WT. Moreover, is lipidation of GABARAP required? A GABARAP variant lacking the C-terminal glycine residue should also be included in Figure 5E and 5F to address this.
- 5) To corroborate that ARMS is targeted to autophagic degradation, the authors should show that ARMS localizes with autophagosome and autolysosomal markers (e.g. p62, LC3B, LAMP1). In addition, the authors should show biochemically that ARMS reside inside of autophagosomes (e.g. by protease protection assays).
- 6) Figure 6C: The authors should show that overexpression of GABARAP L63Q does not affect the levels of ARMS EE. As an additional negative control, the authors should overexpress LC3B.
- 7) While ARMS increases slightly upon 6 h CQ treatment in Figure 6E, this does not seem to be the case in Figure 6G (in the absence of GABARAP overexpression). This raises concerns about the robustness of this phenotype.
- 8) What is the effect of GABARAP KD or KO on ARMS protein levels? Does ARMS increase in a similar extent as seen with CQ treatment? Similarly, the authors should assess the effect of ATG7 inhibition.
- 9) Importantly, it is not clear under which physiological conditions ARMS level would increase to the point that GABARAP would antagonize this. Are elevated ARMS levels actually bad for neurons?

Referee #3:

Synopsis

In their manuscript, Jiang et al present a set of experiments that identify a direct interaction between the 220 kDa neuronal scaffold protein ARMS (aka KIDINS220) and the ubiquitin-like proteins of the GABARAP subfamily of ATG8/LC3/GABARAP proteins associated with membrane scaffolding, trafficking, and autophagy. They determine a crystal structure of the ARMS:GABARAP complex revealing a novel type of interaction, which involves hydrophobic pockets (HPs) 1 and 2 of the GABARAP but a novel sequence within the ARMS ankyrin repeat region. This is different from the LC3-Interacting Region (LIR)-type of interaction described for other proteins. Using biochemical assays, Jiang et al identify and confirm ARMS and GABARAP mutants, which disrupt the 1:1 complex formation between ARMS and GABARAP, thus offering tools to study the role of the interaction in the cellular context. ARMS is key for dendritic spine formation and neuronal function, and perturbation in its abundance and/or localization have an impact on neuronal biology. The authors utilize the dendritic spine formation assays in hippocampal neurons to study the of ARMS:GABARAP complex formation or disruption. Thus, mutants or peptides that abolish ARMS:GABARAP binding also impact spine formation emphasizing the role of this complex in ARMS stabilization and localization in neurons and more broadly in neuronal biology.

This study attempts to bring significant novelty in molecular biology and deserves interest. However, there are some important improvements needed to ensure the study reaches the right quality for being accepted by the journal. I am offering my comments below to stimulate submission of an improved manuscript.

Specific comments

- Protein & domain identity: UniProt ID Q9EQG6 belongs to rat ARMS/KIDINS220 and NOT rabbit as stated in the methods. Further, according to the database, the rat version contains 11x ANK repeats and NOT 13x (as indicated by the authors on Fig. 1)! Authors therefore should clarify:

(1) Was rat, rabbit, or human version of the ARMS protein studied? Please show the entire sequence (best with a multiple interspecies alignment) in the manuscript. Define every ANK repeat in the sequence. Also indicate other relevant domains, including the putative LIR motifs and the newly identified GABARAP-binding sequence. Correct the figure 1 by drawing the right number of ankyrin repeats.

(2) How many ankyrin repeats does this studied construct contain and which exactly ankyrin repeat position is studied?

(3) I would assume the human version of ARMS with predicted 12x ANK repeats (UniProt Q9ULH0) was not studied as the hippocampal neurons were from the rat and the authors wanted to be consistent with the sequences of the endogenous proteins. Can the authors confirm this rationale?

- ARMS binds GABARAPs and not LC3s: It is good to see results of the binding studies for all 6 LC3/GABARAP homologs. For a good overview, however, it would be of advantage to also have a summary table with Kd values for the LC3/GABARAP family for clarity.

- ARMS:GABARAP complex and novel GABARAP-binding motif E36/E67-L69-D70: I am asking the authors to highlight the novel GABARAP-binding sequence on the full-length multiple alignment of ARMS proteins in various species to show evolutionary conservation (which I believe is given between human and rat) and compare to putative LIRs clearly. Any conservation in other ankyrin repeats of other proteins? Please add to the Discussion section how representative the new interaction surface may be in the realm of ankyrin repeat proteins, especially in the light of the previously published LIR-based sequence in another neuronal protein ANK3/Ankyrin G (Tseng et al 2014 PNAS "Giant ankyrin-G stabilizes somatodendritic GABAergic synapses through opposing endocytosis of GABAA receptors" and Li et al 2018 Nat Chem "Biol Potent and specific Atg8-targeting autophagy inhibitory peptides from giant ankyrins").

- Use of ARMS overexpression to study its interaction with GABARAP and role of autophagy in regulating ARMS abundance. Overexpression of ARMS in HEK293T to study interaction with GABARAP by Co-IP may NOT be physiological. Too high levels of ARMS might have led to ARMS aggregation and ubiquitylation and the involvement of aggrephagy factors, such as p62/SQSTM1 or NBR1, in its degradation (rather than direct interaction with GABARAP). So, to exclude this artifact, the authors should check if the overexpressed (vs endogenous) ARMS is found prevalently in Triton X-100 insoluble fraction. If aggregation is massive, I would dismiss the results of the Co-IP studying various non-GABARAP-binding mutants and instead focus on the endogenous Co-IP in neurons. Knockout GABARAP and reconstitution with non-ARMS-binding mutants may be a more elegant strategy avoiding protein aggregation. Alternatively, levels of plasmids may need to be varied to achieve protein levels that are close to physiological.

- Role of autophagy in regulating ARMS abundance: As mentioned above, overexpression of ARMS may lead to aggregation and induce autophagy via alternative pathways (ubiquitin-dependent aggrephagy). So, a better approach would be to check if ARMS endogenous levels are changing if GABARAP is knocked out in neurons. Importantly, authors are advised to monitor changes in endogenous p62/SQSTM1 levels in their chloroquine assay as an orthogonal readout for autophagy inhibition. Further, use of ATG7 KO in neurons (perhaps can be obtained from ATG7 KO mouse labs - not rat sequence then) is strongly recommended to demonstrate the effects of autophagy disruption on ARMS levels. This avoids overexpression of ARMS and is a gold standard in the autophagy research. Abundance of p62/SQSTM1 should always be used as a control for intact autophagy. As the authors state that GABARAP influences ARMS abundance only upon overexpression (excess of protein), it is again likely that aggregation may be the reason for the studied partial (and not full) effects of ARMS:GABARAP complex disruption in neurons. Check aggregation potential of ARMS by Triton solubility assay also in neurons.

- Dendritic spine assay: This is an elegant assay and a suitable one to claim physiological effects of the ARMS:GABARAP interaction. However, the key question is whether the EE mutant is disrupting the structure of ARMS ankyrin repeats and in this way destabilizes the protein (making it perhaps proteasomal substrate)? Can the authors check the folding status of the single and double mutants in the purified N-terminal constructs of ARMS. Especially in dendritic spine formation assay, misfolded ARMS could be mistaken for the lack of GABARAP binding. What would happen if non-ARMS binding mutants of GABARAP are co-expressed with ARMS in this assay?

- Competition between ARMS and AnkB WT peptide: It is hard to interpret the result of the ITC experiments where only one sequence of events leads to the displacement of the ARMS:GABARAP binding. Affinity of AnkB peptide is much higher, so in theory it should always displace ARMS from the GABARAP. Perhaps, it would be possible to complement this method of inhibiting ARMS:GABARAP complex formation by reconstitution of mutant forms of GABARAP in neurons. Also, would it be possible to use isolated GABARAP-binding peptides from ARMS itself in the competition assay? Would it be possible to enhance its affinity and make more like AnkB. That would be of great interest for the field, and this point should at least be mentioned in Discussion. Lastly, the GFP construct was used to bring GABARAP-competing peptides in the cells. Would it be possible to use cell-penetrating peptides instead?

Point-by-point responses to the comments raised by the reviewers

Before point-by-point responses to the referees' comments, we thank all the reviewers for recognizing the novelty and interests of our works and their critical and constructive suggestions and guidance that help us to efficiently improve our manuscript. Our responses are shown in blue.

Referee #1:

Jiang et al report a detailed biochemical and structural characterization of the interaction between the scaffolding protein ARMS and GABARAP. They show by solving the crystal structure of a fusion protein of the first 4 ankyrin repeats and GABARAP a new interaction mode that does not rely on any classical LIR peptides. Instead, the two classical hydrophobic pockets of GABARAP are recognized by a network of polar and hydrophobic interactions. The authors validate these by mutagenesis and show that interaction with the LC3 subfamily of ATG8 proteins does not occur unless LC3s get mutated to allow critical interactions that are present in GABARAP. The authors also show that these interactions are functionally important in cellular assays.

Overall, this is an interesting and carefully designed study. My only comment is that it is not quite clear to me how autophagy regulates the protein level. ARMS is a membrane protein and including more information in which membranes it occurs and if autophagy includes removal part of the membranes as well or if a protein retrieval mechanism into the cytoplasm is required. Adding this to the discussion would strengthen the manuscript.

We sincerely thank the reviewer for the positive and encouraging comments, as well as for the insightful suggestion. ARMS is a membrane-spanning scaffold protein primarily localized at the plasma membrane and intracellular vesicular compartments, particularly within neuronal processes and growth cones, where it regulates signaling and trafficking^[1,2]. Previous studies indicate that ARMS can also influence autophagy through the NF- κ B–ROS pathway, leading to autophagic cell death under stress^[3]. However, how autophagy directly regulates ARMS abundance remains unclear. As the reviewer pointed out, membrane-associated proteins can be targeted by autophagy through at least two routes: (i) engulfment of membrane portions containing the cargo, or (ii) internalization or membrane retrieval followed by cytosolic recognition by ATG8 proteins. Given that ARMS dynamically cycles between the plasma membrane and endosomal compartments, it is plausible that ARMS is first retrieved from the membrane before being recognized by GABARAP for selective degradation. Alternatively, ARMS may be incorporated into autophagosomes as part of membrane subdomains. While our current study does not distinguish between these possibilities, the non-canonical GABARAP interaction uncovered here may facilitate such selective recruitment. We have now added this point into the discussion part of the revised manuscript as guided by the reviewer.

Referee #2:

Summary

In this manuscript, Jiang et al. examined how the neuronal scaffold protein ankyrin repeat-rich membrane spanning (ARMS) is regulated by its binding partners. Firstly, the authors found that members of the GABARAP subfamily of ATG8 proteins directly bind to ARMS with low μM affinities. Binding site mapping revealed that this interaction is mediated by the first four ankyrin repeats of ARMS (1-4 ARMS ARs). Surprisingly, the crystal structure of ARMS binding region in complex with GABARAP showed that the first two finger-like loops connecting adjacent ankyrin repeats insert into GABARAP's hydrophobic pockets (HP1, HP2) via prominent electrostatic interactions. Intriguingly, key residues involved in these interactions are absent in LC3 members of the ATG8 family, explaining the lack of interaction with LC3 proteins. Next, the authors sought to assess the significance of the ARMS-GABARAP binding in neuronal cells and found that rescue of ARMS knockdown (KD) with a GABARAP-binding deficient variant (EE) fails to restore dendritic growth and spine density to the level of wild-type ARMS. Lastly, the authors showed that GABARAP-binding to ARMS controls its abundance. Collectively, the work of Jiang and colleagues clearly establish ARMS as new ATG8 interacting protein and provide evidence for an exciting new LIR independent, non-canonical binding mode. However, the functional relevance of this interaction is less clear. Several critical points remain to be addressed to strengthen this part of the study.

We sincerely thank the reviewer for the thoughtful and encouraging summary of our work. We are pleased that the reviewer finds our biochemical and structural analysis of the ARMS–GABARAP interaction to be novel and rigorous, and appreciates the discovery of a non-canonical, LIR-independent binding mode. At the same time, we fully acknowledge the reviewer's concerns regarding the physiological relevance and functional implications of this interaction. In the revised manuscript, we have carefully addressed each of the specific points raised below through additional data, new experiments, and clarifying discussion, as detailed in the point-by-point responses.

1) Does ARMS interact with GABARAP in neurons? It is not evident how 293T cells help to assess the ARMS-GABARAP interaction in a physiological context (Figure 4A).

We thank the reviewer for raising this important point regarding the physiological relevance of the ARMS–GABARAP interaction. While our initial co-immunoprecipitation assays were conducted in HEK293T cells to establish biochemical binding, we fully agree that validating the interaction in a neuronal context is essential to support its functional implications. It is worth noting that both ARMS and GABARAP are endogenously expressed not only in the nervous system, but also in other tissues, such as kidney, where they are known to play important physiological roles^[4,5]. Nonetheless, given that our study focuses primarily on the role of ARMS in dendritic morphogenesis and synaptic development, we share the reviewer's view that direct demonstration of their interaction in neurons is particularly important.

To address this, we initially attempted co-immunoprecipitation in primary hippocampal neurons

(DIV7) co-transfected with Flag-ARMS and GFP-GABARAP. However, due to the low transfection efficiency in neurons and the large size of ARMS, Flag-ARMS expression was undetectable by Western blot. We therefore shifted to an endogenous approach and successfully performed co-immunoprecipitation in DIV7 hippocampal neurons transfected with GFP-GABARAP or GFP. Using anti-GFP beads followed by anti-ARMS immunoblotting, we detected a robust interaction between endogenous ARMS and GFP-GABARAP (Figure R1), confirming that this complex indeed forms under physiological neuronal conditions. We have now incorporated these new results into the revised manuscript as guided by the reviewer.

Figure R1. Endogenous ARMS interacts with GFP-GABARAP in primary hippocampal neurons. Primary hippocampal neurons were transfected at DIV5 with either GFP or GFP-GABARAP. At DIV7, cell lysates were subjected to co-immunoprecipitation using anti-GFP beads, followed by immunoblotting with anti-ARMS antibody. A specific interaction between endogenous ARMS and GFP-GABARAP was detected, indicating that the ARMS-GABARAP complex forms under neuronal conditions.

2) Does KD of GABARAP phenocopy the effect of ARMS EE re-expression in ARMS KD with regard to dendritic growth and spine density (Figure 4B and 4D)? In other words, what happens to dendrites and spines when GABARAP is depleted during ARMS WT re-expression? Along similar lines, the authors should assess the effect of ATG7 inhibition. Does this prevent the effect of ARMS WT re-expression?

We thank the reviewer for this thoughtful and constructive comment. In our original data, ARMS WT, but not the GABARAP-binding-deficient EE mutant, restored dendritic and synaptic phenotypes in ARMS knockdown neurons (Figure 4). To address the functional requirement of GABARAP and autophagy in ARMS-mediated neuronal development, we performed additional experiments analyzing dendritic arborization and dendritic spine formation under conditions of GABARAP or ATG7 knockdown.

In early-stage neurons (transfected at DIV4 and analyzed at DIV7), ARMS knockdown impaired dendritic arborization, and this phenotype was fully rescued by ARMS WT but not the GABARAP-binding-deficient mutant ARMS EE. Importantly, even under GABARAP or ATG7 knockdown, ARMS WT retained its ability to rescue dendritic arborization, whereas ARMS EE remained non-functional (Figure R2A-B). These results suggest that during early morphogenesis, ARMS-mediated dendritic outgrowth does not strictly depend on GABARAP or autophagy,

possibly due to functional compensation by other Atg8 family members or alternative mechanisms.

In contrast, in later-stage neurons (transfected at DIV10 and analyzed at DIV14), ARMS WT failed to rescue dendritic spine density or maturation under either GABARAP or ATG7 knockdown conditions, and ARMS EE again had no effect (Figure R2C–D). Together, these findings reveal that ARMS function in neuronal morphogenesis shows context-dependent reliance on GABARAP binding and autophagy, with a more pronounced requirement during later stages of synaptic maturation. The precise mechanisms underlying this differential sensitivity remain to be fully elucidated and warrant further investigation. We have integrated these results and their mechanistic implications into the revised manuscript as guided by the reviewer.

Figure R2. Knockdown of GABARAP or ATG7 impairs ARMS-mediated neuronal development. (A) Representative confocal images of hippocampal neurons transfected at day 4 with BFP-ARMS-shRNA, and co-transfected with one of the following: GFP and RFP-ARMS WT; GFP-GABARAP-shRNA and RFP-ARMS WT or RFP-ARMS EE; GFP-ATG7-shRNA and RFP-ARMS WT or RFP-ARMS EE. Neurons were fixed and imaged at day 7. ARMS WT fully rescued dendritic arborization even under GABARAP or ATG7 knockdown, whereas the EE mutant failed to do so. Scale bar, 30 µm.

(B) Quantification of neurite length in neurons from panel A. ARMS WT significantly restored neurite growth, including under GABARAP or ATG7 knockdown, while ARMS EE showed no

rescue. Error bars represent SEM. ****P<0.0001, ns: not significant.

(C) Representative images of hippocampal neurons transfected at day 10 with BFP-ARMS-shRNA, and co-transfected with the same five experimental conditions as in panel A. Neurons were fixed and imaged at day 14. ARMS WT failed to rescue dendritic spine density under GABARAP or ATG7 knockdown, and ARMS EE showed no rescue in all conditions. Upper panels: low magnification (scale bar, 50 μ m); lower panels: high magnification of dendritic spines (scale bar, 5 μ m).

(D) Quantification of dendritic spine density under the five experimental conditions described in panel C. ARMS WT failed to restore spine density under GABARAP or ATG7 knockdown, similar to the non-rescuing EE mutant. n = 8–11 neurons from three independent cultures. Error bars indicate SEM. ****P<0.0001, ns: not significant.

3) The authors need to show by Western blot that ARMS WT and EE are expressed to similar levels in these neurons (Figure 4B and 4D). The same applies to Figure 5 and GFP and GFP-GABARAP as well as to Figure 7F and GFP-AnkB WT and WR.

We thank the reviewer for this important comment. To assess the relative expression levels of the constructs used in Figure 4B, 4D, 5A, 5D, and 7F, we performed parallel transfections in primary hippocampal neurons under identical conditions. We first examined the expression levels of RFP-tagged ARMS WT and EE mutants. Due to low transfection efficiency of these large constructs in neurons, Western blot (WB) signals were undetectable despite multiple attempts. We therefore repeated the transfection in HEK293T and HeLa cells using the same constructs. WB results confirmed that RFP-ARMS WT and RFP-ARMS EE were expressed at comparable levels in both cell lines (Figure R3A–B). For constructs used in the updated Figure 5, we transfected neurons with GFP, GFP-GABARAP WT, or GFP-GABARAP mutants (L63Q and G116A). WB analysis confirmed similar expression levels across all constructs (Figure R3C–D).

In contrast, for constructs used in Figure 7F, Western blot analysis showed that GFP-AnkB WR exhibited significantly lower total expression in neurons compared to GFP-AnkB WT (Figure R3E–F). To investigate the underlying cause, we transfected GFP-AnkB WT and WR constructs into HEK293T cells and quantified the ratio of GFP-positive cells. This revealed that the transfection efficiency of the WR construct was markedly lower than that of WT (Figure R3G–H), which explains the reduced protein levels observed in the neuronal lysates. Importantly, in fixed-cell imaging experiments (Figure 7), both constructs exhibited comparable fluorescence intensity within transfected neurons. These results indicate that the observed decrease in total WR expression in WB is due to lower transfection efficiency across the population, rather than a reduction in expression within individual neurons. Therefore, this difference does not compromise the interpretation of WR as a functional mutant control in our assays.

Together, these results validate that the expression levels of most constructs used in functional assays were comparable. The observed expression difference in AnkB WR is due to lower transfection efficiency, rather than intrinsic instability or degradation. We have integrated these results into the revised manuscript (updated Figure EV4) as guided by the reviewer.

Figure R3. Western blot analysis of construct expression levels in transfected cells.

(A–B) RFP-ARMS WT or EE constructs were transfected into HEK293T and HeLa cells. Western blot analysis showed similar expression levels (A), and quantification from three independent replicates confirmed no significant difference (B). Data represent mean \pm SEM; ns: not significant.

(C–D) GFP, GFP-GABARAP WT and mutants (L63Q or G116A) were transfected into primary hippocampal neurons. Western blot analysis (C) and quantification (D) showed comparable expression levels across all constructs ($n = 3$). Data represent mean \pm SEM; ns: not significant.

(E–F) GFP-AnkB WT or WR were transfected into neurons. Western blot analysis revealed significantly reduced expression of the WR construct (E). Quantification confirmed the difference in protein levels (F). Data represent mean \pm SEM; **** $P < 0.0001$, ns: not significant.

(G–H) To assess whether reduced WR expression was due to lower transfection efficiency, GFP-AnkB WT and WR were transfected into HEK293T cells. Fluorescence-based quantification of GFP-positive cells (G) showed significantly lower transfection efficiency for WR (H),

explaining the reduced protein levels observed in neurons. Data represent mean \pm SEM; ****P<0.0001.

4) *What is the effect of ARMS-binding deficient GABARAP variant (e.g. L63Q) expression on spine density and maturation? The authors should expand Figure 5E and 5F by including GABARAP L63Q and compare it to GABARAP WT. Moreover, is lipidation of GABARAP required? A GABARAP variant lacking the C-terminal glycine residue should also be included in Figure 5E and 5F to address this.*

We thank the reviewer for raising this important point. Guided by this suggestion, we expanded our analysis to directly test whether the ARMS–GABARAP interaction and GABARAP lipidation are functionally required for dendritic spine regulation. Specifically, we co-expressed RFP-ARMS with either the ARMS-binding deficient GABARAP mutant (L63Q) or the lipidation-defective G116A mutant in primary hippocampal neurons at DIV10 and performed immunostaining at DIV18 to evaluate spine number and morphology.

As previously shown (Figure 5D–F), overexpression of GABARAP WT inhibited the ARMS-induced increase in spine density and maturation, while co-expression of ARMS EE with GABARAP partially rescued spine number but did not improve spine maturity. In the newly added experiments, we found that GABARAP L63Q, which is deficient in ARMS binding, largely relieved the suppressive effect of GABARAP on both spine number and maturation (Figure R4A–C). This result strongly suggests that physical interaction with ARMS is required for GABARAP to modulate ARMS-dependent spine development. In contrast, co-expression of the G116A mutant, which cannot undergo lipidation and membrane association, led to a dramatic reduction in both spine number and the proportion of mature spines, even more severe than that caused by GABARAP WT (Figure R4A–C). These observations indicate that GABARAP lipidation and autophagy-related functions are critically involved in synaptic development.

Together, these results reinforce our model that GABARAP negatively regulates ARMS-mediated synaptic maturation through both direct binding and its membrane-associated functions. They also suggest that proper coordination between scaffold protein interactions and autophagic pathways is essential for dendritic spine development. While the precise mechanism linking GABARAP lipidation to spine remodeling remains to be further elucidated, these findings highlight a potentially important avenue for future investigation. We have integrated these results into the revised manuscript (updated Figure 5) as guided by the reviewer.

Figure R4. ARMS–GABARAP interaction and GABARAP lipidation regulate dendritic spine development and maturation.

(A) Representative images of hippocampal neurons co-transfected at day 10 with RFP-ARMS and either GFP-GABARAP, GFP-GABARAP L63Q, or GFP-GABARAP G116A. Neurons were fixed and immunostained at day 18. Upper panels show low-magnification views with regions of interest marked by white boxes (scale bar, 50 μ m); lower panels show high-magnification images of dendritic spines (scale bar, 5 μ m).

(B-C) Quantification of dendritic spine density (B) and the proportion of mature mushroom-shaped spines (C) under each condition. Co-expression of GABARAP L63Q with ARMS significantly restored spine density and maturation compared to GABARAP WT, indicating loss of inhibitory interaction. In contrast, co-expression of GABARAP G116A led to a marked reduction in both spine number and maturity. GFP-GABARAP and RFP-ARMS, n=6; GFP-GABARAP L63Q and RFP-ARMS, n=8; GABARAP G116A and RFP-ARMS, n=10. Data represent mean \pm SEM; ****P<0.0001.

5) *To corroborate that ARMS is targeted to autophagic degradation, the authors should show that ARMS localizes with autophagosome and autolysosomal markers (e.g. p62, LC3B, LAMP1). In addition, the authors should show biochemically that ARMS reside inside of autophagosomes (e.g. by protease protection assays).*

We thank the reviewer for raising this insightful question regarding the autophagic fate of ARMS. To address whether ARMS is targeted for autophagic degradation, we examined its localization relative to autophagosome and autolysosome markers under autophagy-inducing conditions.

Guided by the reviewer's suggestion, we overexpressed RFP-ARMS in HEK293T cells and treated them with EBSS to induce autophagy. In a time-course assay, we observed that 3-hour EBSS treatment robustly induced the formation of LC3B-positive puncta (Figure R5A), establishing this as a reliable timepoint for downstream analysis. Under these conditions, immunostaining for the lysosomal marker LAMP1 revealed a marked increase in its expression and extensive colocalization with RFP-ARMS (Figure R5B). Notably, in control cells lacking ARMS expression, LAMP1 fluorescence was weak or nearly undetectable, suggesting that overexpression of ARMS may increase lysosomal load and promote lysosomal activation during autophagy. Although LAMP1 did not form typical puncta in these conditions, we frequently observed elongated or tubular lysosomal structures, consistent with enhanced lysosomal accumulation. This phenotype likely reflects active autophagic flux in cells overexpressing ARMS. In parallel, we co-expressed RFP-ARMS with GFP-LC3B and found that EBSS treatment promoted the formation of large LC3B-positive aggregates that strongly colocalized with ARMS (Figure R5C). Together, these results indicate that ARMS localizes to both autophagosomes and lysosomes upon autophagy induction, supporting the notion that ARMS is subject to autophagic regulation.

Next, we performed protease protection assays to biochemically confirm whether ARMS resides within autophagosomal structures as guided by the reviewer. HEK293T cells overexpressing Flag-ARMS were treated with chloroquine to accumulate autophagosomes, which were subsequently isolated by gradient centrifugation and divided into four aliquots for differential treatments. The first aliquot (with protease inhibitor, unboiled) showed no detectable ARMS, p62, or LC3 bands, confirming the absence of cytosolic contamination in the crude autophagosome preparation. The second aliquot (with protease inhibitor, boiled) displayed strong signals for Flag-ARMS, p62, and LC3 II, validating successful enrichment of intact autophagosomes. In contrast, the third aliquot (treated with proteinase K alone) exhibited identical band patterns to lane 2, indicating that ARMS was protected from proteolytic digestion and localized inside membrane-enclosed compartments. Finally, addition of Triton X-100 together with proteinase K led to complete degradation of ARMS, p62, and LC3, demonstrating that detergent permeabilization disrupted the vesicle membrane barrier (Figure R6). Collectively, these results confirm that ARMS resides within autophagosomes, protected from proteolysis by the surrounding double-membrane structure (Figure R6).

Figure R5. EBSS-induced autophagy promotes colocalization of ARMS with autophagosomal and lysosomal markers.

(A) GFP-LC3B was expressed in HEK293T cells, followed by autophagy induction using EBSS for 0.5 h, 1 h, or 3 h. Confocal imaging revealed prominent puncta formation by GFP-LC3B after 3 h of treatment, indicating accumulation of autophagosomes (scale bar, 10 μ m).

(B) RFP-ARMS was overexpressed in HEK293T cells, which were then treated with EBSS for 3 h and immunostained for the lysosomal marker LAMP1. EBSS induction led to elevated LAMP1 levels and strong colocalization with ARMS (scale bar, 5 μ m).

(C) Co-expression of RFP-ARMS and GFP-LC3B in HEK293T cells followed by EBSS treatment for 3 h resulted in colocalized punctate aggregates of ARMS and LC3B, supporting their convergence within the autophagic pathway (scale bar, 5 μ m).

Figure R6. Protease protection assay demonstrates that ARMS localizes within membrane-enclosed autophagosomes.

HEK293T cells transfected with Flag-ARMS were treated with chloroquine for 10 h to accumulate autophagosomes. Crude autophagosomes were isolated by gradient centrifugation and divided into four aliquots for differential treatment prior to WB analysis: Lane 1: Protease inhibitor only, unboiled; Lane 2: Protease inhibitor only, boiled; Lane 3: Proteinase K only, boiled; Lane 4: Proteinase K plus Triton X-100, boiled. These results confirm that ARMS is sequestered inside autophagosomes and shielded from degradation unless vesicle membranes are disrupted.

6) *Figure 6C: The authors should show that overexpression of GABARAP L63Q does not affect the levels of ARMS EE. As an additional negative control, the authors should overexpress LC3B.*

We thank the reviewer for this important suggestion. To further examine the specificity of GABARAP-mediated regulation of ARMS, we performed Western blot analysis following co-transfection of Flag-ARMS or Flag-ARMS EE with GFP, GFP-GABARAP, GFP-GABARAP L63Q, or GFP-LC3B in HEK293T cells. As shown in Figure R7A–B, co-expression of GABARAP with ARMS led to a significant reduction in ARMS protein levels, consistent with our previous findings. In contrast, the GABARAP L63Q mutant, which disrupts ARMS binding, failed to reduce ARMS levels, suggesting that direct interaction is required for this regulation. Importantly, GABARAP did not alter the expression of the ARMS EE mutant, further supporting that this mutant is resistant to GABARAP-mediated degradation. Additionally, co-expression with LC3B, a closely related Atg8 family member that does not interact with ARMS, had no effect on either ARMS WT or EE levels (Figure R7). These results confirm that GABARAP specifically promotes the downregulation of ARMS through a binding-dependent mechanism, and that neither LC3B nor the L63Q mutant can substitute this function. We have integrated these results into the revised manuscript (updated Figure 6) as guided by the reviewer.

Figure R7. GABARAP, but not its L63Q mutant or LC3B, reduces abnormal accumulation of ARMS levels in cells.

(A) HEK293T cells were co-transfected with Flag-ARMS WT or Flag-ARMS EE together with GFP, GFP-GABARAP, GFP-GABARAP L63Q, or GFP-LC3B as indicated. Western blot analysis showed that co-expression of GABARAP markedly reduced ARMS WT protein levels, whereas the L63Q mutant and LC3B had no significant effect. Notably, GABARAP did not reduce the levels of the ARMS EE mutant.

(B) Quantification of Flag-ARMS band intensity normalized to GFP from three independent experiments. Data represent mean \pm SEM; ** $P < 0.01$; ns, not significant.

7) While ARMS increases slightly upon 6 h CQ treatment in Figure 6E, this does not seem to be the case in Figure 6G (in the absence of GABARAP overexpression). This raises concerns about the robustness of this phenotype.

We thank the reviewer for this important observation. As noted, the CQ-induced increase in ARMS levels is modest in Figure 6E and not clearly recapitulated in Figure 6G under conditions without GABARAP overexpression, raising concerns about reproducibility. We agree that the phenotype appears variable across experimental contexts and appreciate the opportunity to clarify this.

We would like to point out that Figures 6E and 6G monitor different forms of ARMS: Figure 6E assesses endogenous ARMS levels, while Figure 6G involves overexpression of Flag-tagged ARMS. The discrepancy may stem from differences in expression level, stability, or post-translational modifications between endogenous and overexpressed ARMS. In particular, high levels of overexpressed ARMS may partially saturate or escape the normal autophagic machinery, resulting in attenuated responsiveness to autophagy inhibition by CQ. Moreover, the extent of ARMS accumulation upon CQ treatment is relatively modest, and appears sensitive to culture conditions, baseline protein levels, and assay timing. We have observed this variability in replicate experiments and now acknowledge it explicitly in the revised Discussion section. Importantly, despite this variability, multiple lines of evidence across our study consistently support the model that ARMS is a target of GABARAP-dependent autophagic regulation: including CQ-induced ARMS accumulation (Figure 6C, D), suppression of ARMS levels by GABARAP but not LC3B or GABARAP mutants (Figure R7), colocalization with LAMP1 and

LC3B upon autophagy induction (Figure R5), and protection of ARMS within isolated autophagosomes (Figure R6). Taken together, these results indicate that ARMS is regulated by autophagy, although this regulation may involve multiple steps and exhibit context-dependent robustness.

8) *What is the effect of GABARAP KD or KO on ARMS protein levels? Does ARMS increase in a similar extent as seen with CQ treatment? Similarly, the authors should assess the effect of ATG7 inhibition.*

We thank the reviewer for this important suggestion. To determine whether inhibition of autophagy-related proteins affects ARMS levels, we performed lentivirus-mediated knockdown of *GABARAP* and *ATG7* in HEK293T cells and examined endogenous ARMS expression by Western blot. The results showed that knockdown of either *GABARAP* or *ATG7* led to a significant accumulation of endogenous ARMS protein compared to control (Figure R8). This phenotype is consistent with the moderate ARMS increase observed under chloroquine (CQ) treatment (updated Figure 6C, D), further supporting that ARMS is subject to autophagic degradation. Importantly, these findings validate the requirement of both *GABARAP*, a core Atg8 family member, and *ATG7*, an essential E1-like enzyme in autophagosome biogenesis, for maintaining normal ARMS homeostasis. Together with our overexpression and colocalization data, these knockdown results strengthen the conclusion that ARMS is regulated via the canonical autophagy pathway. We have integrated these results into the revised manuscript (updated Figure 6) as guided by the reviewer.

Figure R8. Knockdown of *GABARAP* and *ATG7* increases endogenous ARMS levels in HEK293T cells.

(A–B) Lentiviral shRNA-mediated knockdown of *GABARAP* was performed in HEK293T cells. Western blot analysis revealed a marked increase in endogenous ARMS protein compared to control (A). Quantification from four independent experiments confirmed statistical significance (B, $n = 4$, mean \pm SEM, **** $P < 0.0001$).

(C–D) Knockdown of *ATG7* was similarly achieved via lentiviral shRNA. Western blot analysis showed a robust increase in ARMS protein level (C), with quantification indicating strong statistical significance (D, $n = 4$, mean \pm SEM, **** $P < 0.0001$).

9) *Importantly, it is not clear under which physiological conditions ARMS level would increase to the point that GABARAP would antagonize this. Are elevated ARMS levels actually bad for neurons?*

We thank the reviewer for this insightful and important question regarding the physiological relevance and functional consequences of *GABARAP*-mediated ARMS regulation. Guided by this

concern, we carefully reviewed our experimental data and relevant literature to clarify under what conditions ARMS becomes subject to GABARAP-dependent degradation, and whether its overaccumulation is detrimental to neuronal function.

Our results show that under basal conditions, overexpression of GABARAP does not alter the endogenous ARMS protein level (updated Figure EV5A, B), suggesting that GABARAP does not actively degrade ARMS when its abundance remains within physiological limits. However, upon artificial overexpression of ARMS, GABARAP promotes its degradation via an autophagy-dependent pathway (updated Figure 6A, B). This indicates that GABARAP functions as a selective quality control factor rather than a general regulator of ARMS homeostasis. This mode of regulation is conceptually similar to how autophagy-related proteins recognize and clear aggregation-prone or dosage-sensitive substrates that disrupt cellular balance when overexpressed.

Given the scaffold nature of ARMS and its multiple interaction domains, aberrantly high expression may lead to inappropriate clustering of signaling complexes or interfere with intracellular trafficking and signaling fidelity. Indeed, ARMS (also known as Kidins220) is known to respond to various environmental cues and serve as a central hub for protein–protein interactions that regulate neuronal survival, differentiation, synaptic plasticity, and dendritic development^[6-8]. Undoubtedly, the normal exertion of ARMS function at physiological levels is indispensable for the maintenance of neuronal homeostasis. Our own functional assays demonstrate that ARMS overexpression markedly increases dendritic spine density and enhances spine maturity, phenotypes often associated with neuronal hyperexcitability. Such structural alterations may disrupt synaptic input integration and perturb neuronal circuit stability. These findings are supported by prior reports showing that early ARMS overexpression impairs neuronal polarization and dendritic growth in cultured hippocampal neurons^[9].

Notably, ARMS dysregulation has been implicated in human disease contexts. In Alzheimer's disease brain tissues, ARMS levels are significantly elevated and show partial colocalization with phosphorylated tau, suggesting a potential involvement in AD pathogenesis^[10]. Moreover, upregulation of ARMS has been reported in multiple cancers, including melanoma and glioblastoma, where it promotes proliferation and survival^[11-12]. Together, these data and observations underscore that tight regulation of ARMS expression is crucial for neuronal and cellular homeostasis, and that GABARAP-mediated degradation may serve as a protective mechanism to limit ARMS accumulation under stress or disease-related conditions. We have now incorporated this interpretation into the discussion part of the revised manuscript to clarify the physiological significance of GABARAP-mediated ARMS regulation as guided by the reviewer.

Referee #3:

Synopsis

In their manuscript, Jiang et al present a set of experiments that identify a direct interaction between the 220 kDa neuronal scaffold protein ARMS (aka KIDINS220) and the ubiquitin-like proteins of the GABARAP subfamily of ATG8/LC3/GABARAP proteins associated with membrane scaffolding, trafficking, and autophagy. They determine a crystal structure of the ARMS:GABARAP complex revealing a novel type of interaction, which involves hydrophobic pockets (HPs) 1 and 2 of the GABARAP but a novel sequence within the ARMS ankyrin repeat region. This is different from the LC3-Interacting Region (LIR)-type of interaction described for other proteins. Using biochemical assays, Jiang et al identify and confirm ARMS and GABARAP mutants, which disrupt the 1:1 complex formation between ARMS and GABARAP, thus offering tools to study the role of the interaction in the cellular context. ARMS is key for dendritic spine formation and neuronal function, and perturbation in its abundance and/or localization have an impact on neuronal biology. The authors utilize the dendritic spine formation assays in hippocampal neurons to study the of ARMS:GABARAP complex formation or disruption. Thus, mutants or peptides that abolish ARMS:GABARAP binding also impact spine formation emphasizing the role of this complex in ARMS stabilization and localization in neurons and more broadly in neuronal biology.

This study attempts to bring significant novelty in molecular biology and deserves interest. However, there are some important improvements needed to ensure the study reaches the right quality for being accepted by the journal. I am offering my comments below to stimulate submission of an improved manuscript.

We sincerely thank the reviewer for the thoughtful and encouraging evaluation of our work. We are pleased that the reviewer recognizes the novelty of the ARMS–GABARAP interaction, the mechanistic insights from our structural and biochemical analyses, and the functional relevance of this complex to dendritic spine development and neuronal physiology. We appreciate the constructive comments aimed at further improving the clarity, rigor, and overall quality of the manuscript. In the revised version, we have carefully addressed each of the reviewer’s concerns, including new experimental validations and textual clarifications. We believe these revisions significantly strengthen the study and more clearly establish the role of GABARAP in regulating ARMS function and neuronal morphology.

Specific comments

1 • Protein & domain identity: UniProt ID Q9EQG6 belongs to rat ARMS/KIDINS220 and NOT rabbit as stated in the methods. Further, according to the database, the rat version contains 11x ANK repeats and NOT 13x (as indicated by the authors on Fig. 1)! Authors therefore should clarify:

(1) Was rat, rabbit, or human version of the ARMS protein studied? Please show the entire sequence (best with a multiple interspecies alignment) in the manuscript. Define every ANK repeat in the sequence. Also indicate other relevant domains, including the putative LIR motifs and the

newly identified GABARAP-binding sequence. Correct the figure 1 by drawing the right number of ankyrin repeats.

(2) How many ankyrin repeats does this studied construct contain and which exactly ankyrin repeat position is studied?

(3) I would assume the human version of ARMS with predicted 12x ANK repeats (UniProt Q9ULH0) was not studied as the hippocampal neurons were from the rat and the authors wanted to be consistent with the sequences of the endogenous proteins. Can the authors confirm this rationale?

We thank the reviewer for this detailed and important comment regarding protein annotation, ankyrin repeat boundaries, and species consistency. We have carefully clarified the following points:

(1) Protein species, sequence identity, and domain annotation:

The ARMS construct used in this study is derived from *Rattus norvegicus* (rat), consistent with the source of primary hippocampal neurons used in our functional assays. The UniProt ID is Q9EQG6, and we apologize for the erroneous listing of “rabbit” in the original Methods section. This has been corrected throughout the revised manuscript.

To clarify the domain organization and species conservation relevant to our study, we performed a multiple sequence alignment of the N-terminal cytoplasmic region (residues 1–500) of ARMS/KIDINS220 across *Homo sapiens* (human), *Rattus norvegicus* (rat), *Mus musculus* (mouse), *Cavia porcellus* (guinea pig), and *Danio rerio* (zebrafish). This region includes the ankyrin repeat domain and the adjacent linker sequence, which encompasses all residues involved in GABARAP interaction. We focused our alignment analysis on this N-terminal segment rather than the full-length ARMS protein (~1762 residues), for two key reasons: 1) the GABARAP-binding interface resides entirely within the N-terminal domain analyzed in our study; 2) a full-length alignment would be excessively long and graphically unwieldy, while offering no additional mechanistic insight.

The alignment confirms high sequence conservation of the ankyrin repeats and adjacent regions across species (Figure R9). We manually defined all ankyrin repeats (ARs 1–13) in the rat ARMS sequence, spanning residues 1–426. These annotations are based on predictions from SMART and PsiPred, as well as manual curation informed by our group’s extensive experience in ankyrin-repeat structure determination^[13]. In Figure R9, we also annotate the previously predicted LIR-like motif, as well as the newly identified GABARAP-binding residues. While early studies typically assigned only 11 ankyrin repeats to ARMS, our reannotation—guided by structural knowledge which includes our previous resolution of the full 24 ANK repeats of ankyrin family structure^[13]—supports the presence of 13 repeats, including divergent N-terminal and C-terminal “capping” repeats. Such terminal repeats often deviate from canonical ankyrin motifs and are prone to being missed by automated domain-calling tools. Similar underannotation has been observed in other ANK-containing proteins, further supporting our curated domain map. In conclusion, our sequence and structural analysis supports the presence of 13 ankyrin repeats in the N-terminal region of rat ARMS, and this revised annotation is now reflected in Figure R9 and

throughout the manuscript.

Figure R9. Sequence alignment of the N-terminal region (residues 1–500) of ARMS/KIDINS220. Multiple sequence alignment was performed using ARMS/KIDINS220 proteins from Homo sapiens (human), Rattus norvegicus (rat), Mus musculus (mouse), Cavia porcellus (guinea pig), and Danio rerio (zebrafish). Absolutely conserved and highly conserved residues are highlighted in dark red and light red, respectively. The predicted LIR-like motifs are marked with green boxes, while the blue stars indicate critical residues identified in this study as essential for binding to GABARAP. The 13 ankyrin repeats (ARs 1–13) are manually defined and labeled. This alignment highlights the strong conservation of the N-terminal cytoplasmic region across species, particularly in the ankyrin repeat domain and GABARAP-interacting interface.

(2) Definition of ankyrin repeat constructs and boundaries used in this study:

To systematically map the GABARAP-binding region within ARMS, we designed and cloned multiple N-terminal fragments of rat ARMS, corresponding to different combinations of ankyrin (ANK) repeats: 1-13 ARs: residues 1–435, encompassing all 13 predicted ANK repeats; 1-8ARs: residues 1–267, encompassing the first 8 ANK repeats; 1-4 ARs: residues 1–130, corresponding to the first 4 ANK repeats; 5-13 ARs: residues 136–435, covering the remaining 9 ANK repeats. These constructs were used in our pull-down mapping experiments (Figure 1D) to localize the GABARAP-binding region. The results clearly demonstrated that the binding activity resides within the N-terminal 1-4 ARs, while the C-terminal 5-13 ARs fragment did not show detectable binding. Based on this mapping, we selected the ARMS 1-4 ARs construct (residues 1–130) for all biochemical binding assays and for crystallization with GABARAP, as it retains full binding activity and is suitable for structural studies. We have updated Figure 1 as guided by the reviewer.

(3) Confirmation of ARMS species used:

We confirm that all experiments in this study used rat ARMS (UniProt Q9EQG6), consistent with the endogenous protein expressed in our primary rat hippocampal neurons. Regarding the UniProt

entry for human ARMS (Q9ULH0) listing only 12 ankyrin repeats, our analysis supports the presence of 13 repeats, as in the rat. This discrepancy likely results from the final capping repeat being missed by automated prediction tools.

We thank the reviewer for prompting this clarification. The revised schematic and alignment (Figure R9) now more accurately depict ARMS domain architecture and species conservation.

2 • ARMS binds GABARAPs and not LC3s: It is good to see results of the binding studies for all 6 LC3/GABARAP homologs. For a good overview, however, it would be of advantage to also have a summary table with K_d values for the LC3/GABARAP family for clarity.

We thank the reviewer for this helpful suggestion. To provide a clear and comprehensive overview of our binding data, we have included a summary table listing the measured dissociation constants (K_d values) for the interaction between ARMS 1–4 ankyrin repeats and all six mammalian Atg8 family proteins (Figure R10). For family members where no binding was detected, we have also indicated the experimental methods used in the manuscript.

Protein	Binding detected with ARMS	K _d (μ M)
GABARAP	Yes	0.9 \pm 0.1
GABARAPL1	Yes	3.8 \pm 1.2
GABARAPL2	Yes	4.6 \pm 1.5
LC3A	No	Not detected
LC3B	No	Not detected
LC3C	No	Not detected

Figure R10. Summary of ARMS 1-4 ARs binding to LC3/GABARAP family proteins.

3 • ARMS:GABARAP complex and novel GABARAP-binding motif E36/E67-L69-D70: I am asking the authors to highlight the novel GABARAP-binding sequence on the full-length multiple alignment of ARMS proteins in various species to show evolutionary conservation (which I believe is given between human and rat) and compare to putative LIRs clearly. Any conservation in other ankyrin repeats of other proteins? Please add to the Discussion section how representative the new interaction surface may be in the realm of ankyrin repeat proteins, especially in the light of the previously published LIR-based sequence in another neuronal protein ANK3/Ankyrin G (Tseng et al 2014 PNAS “Giant ankyrin-G stabilizes somatodendritic GABAergic synapses through opposing endocytosis of GABAA receptors” and Li et al 2018 Nat Chem Biol “Potent and specific Atg8-targeting autophagy inhibitory peptides from giant ankyrins”).

We thank the reviewer for this thoughtful suggestion regarding the evolutionary conservation and broader relevance of the ARMS–GABARAP interaction motif.

As addressed in our response to Comment #1, we have provided a multiple sequence alignment of the N-terminal of ARMS containing all the ankyrin repeats from representative vertebrates, highlighting both the putative LIR motifs and the newly identified GABARAP-binding residues

(E36, E67, L69, D70). This analysis confirms that these key residues are highly conserved across species, including between human and rat (Figure R9).

As in the previous response, we have now included a multiple sequence alignment of ARMS from representative vertebrate species (human, rat, mouse, *Cavia porcellus*, zebrafish), highlighting the novel GABARAP-binding motif in blue box (E36, E67, L69, D70). As shown in newly added Fig R8, this motif is highly conserved across species, supporting its functional importance. While short LIR-like motifs can be found in other ankyrin-repeat proteins, our study shows that the canonical [W/F/Y]xx[L/I/V] motifs in ARMS do not mediate binding to GABARAP. In contrast, the actual GABARAP-binding interface resides within loop regions of the ankyrin repeats and is defined by a unique combination of conserved residues not found in other ankyrin-repeat proteins examined. Ankyrin-G (ANK3), as previously reported^[14, 15], interacts with GABARAP/LC3 through a canonical LIR sequence located outside its ankyrin domain. Structural and sequence analysis indicate that other ankyrin-repeat proteins do not share the conserved residues or loop features observed in ARMS that mediate this interaction, suggesting that the ARMS-GABARAP interface represents a distinct and unique mode of interaction among ankyrin-repeat proteins.

As suggested, we have now added a paragraph to the discussion comparing this noncanonical binding mode with the canonical LIR-mediated interactions previously reported for ANK3/Ankyrin-G.

4• Use of ARMS overexpression to study its interaction with GABARAP and role of autophagy in regulating ARMS abundance. Overexpression of ARMS in HEK293T to study interaction with GABARAP by Co-IP may NOT be physiological. Too high levels of ARMS might have led to ARMS aggregation and ubiquitylation and the involvement of autophagy factors, such as p62/SQSTM1 or NBR1, in its degradation (rather than direct interaction with GABARAP). So, to exclude this artifact, the authors should check if the overexpressed (vs endogenous) ARMS is found prevalently in Triton X-100 insoluble fraction. If aggregation is massive, I would dismiss the results of the Co-IP studying various non-GABARAP-binding mutants and instead focus on the endogenous Co-IP in neurons. Knockout GABARAP and reconstitution with non-ARMS-binding mutants may be a more elegant strategy avoiding protein aggregation. Alternatively, levels of plasmids may need to be varied to achieve protein levels that are close to physiological.

We thank the reviewer for raising this important point regarding potential aggregation artifacts caused by ARMS overexpression in HEK293T cells. To directly address this concern, we performed Triton X-100 solubility fractionation to assess whether ARMS forms detergent-insoluble aggregates. Western blot analysis revealed that both endogenous ARMS (Figure R11A) and overexpressed Flag-ARMS (Figure R11B) are predominantly found in the soluble fraction, with only minimal signal detected in the insoluble pellet. These results indicate that ARMS remains largely soluble under our experimental conditions and does not undergo significant aggregation upon overexpression. Importantly, we also performed GFP-GABARAP pull-down in primary neurons, where endogenous ARMS was clearly detected in the complex without any ARMS overexpression (please see Figure R1 in this response file). This provides

strong evidence that the interaction is physiological and not an artifact of aggregation or bridging adaptors such as p62 or NBR1. Taken together, these results demonstrate that the ARMS–GABARAP interaction is specific and physiological under our experimental conditions, and not driven by overexpression-induced aggregation.

Figure R11. Endogenous and overexpressed ARMS proteins are predominantly found in the Triton X-100-soluble fraction.

(A) Lysed HEK293T cells using lysis buffer containing 1% Triton X-100, followed by Western blot analysis of the resulting supernatant and pellet fractions. The results demonstrated that endogenous ARMS protein was almost exclusively detected in the supernatant.

(B) Overexpressed Flag-ARMS in HEK293T cells, lysed cells using lysis buffer containing 1% Triton X-100, followed by Western blot analysis of the resulting supernatant and pellet fractions. The results demonstrated that Overexpressed ARMS protein was almost detected in the supernatant, with only a faint band observed in the insoluble fraction.

5• Role of autophagy in regulating ARMS abundance: As mentioned above, overexpression of ARMS may lead to aggregation and induce autophagy via alternative pathways (ubiquitin-dependent aggregation). So, a better approach would be to check if ARMS endogenous levels are changing if GABARAP is knocked out in neurons. Importantly, authors are advised to monitor changes in endogenous p62/SQSTM1 levels in their chloroquine assay as an orthogonal readout for autophagy inhibition. Further, use of ATG7 KO in neurons (perhaps can be obtained from ATG7 KO mouse labs - not rat sequence then) is strongly recommended to demonstrate the effects of autophagy disruption on ARMS levels. This avoids overexpression of ARMS and is a gold standard in the autophagy research. Abundance of p62/SQSTM1 should always be used as a control for intact autophagy. As the authors state that GABARAP influences ARMS abundance only upon overexpression (excess of protein), it is again likely that aggregation may be the reason for the studied partial (and not full) effects of ARMS: GABARAP complex disruption in neurons. Check aggregation potential of ARMS by Triton solubility assay also in neurons.

We thank the reviewer for the valuable suggestions on better validating the physiological relevance of ARMS degradation through autophagy. To directly assess whether endogenous ARMS is regulated by autophagy under physiological conditions, we treated primary neurons with a selective ATG7 inhibitor (ATG-IN-2). Western blot analysis showed a clear increase in ARMS

protein levels upon ATG7 inhibition (Figure R12A, B), indicating that ARMS degradation depends on autophagic activity. To validate effective autophagy inhibition, we concurrently examined levels of p62 and LC3. As expected, ATG-IN-2 treatment led to robust accumulation of p62 and increased LC3-I levels, confirming successful disruption of autophagy flux (Figure R12A). These findings are consistent with our previous experiments in HEK293T cells, where knockdown of ATG7 or GABARAP also increased endogenous ARMS levels (please see Figure R8 in this response file), further supporting that ARMS is degraded by a canonical autophagy pathway involving GABARAP.

To evaluate whether aggregation contributes to ARMS degradation, we examined the solubility of endogenous ARMS in neurons by Triton X-100 fractionation. The vast majority of ARMS remained in the soluble fraction under both control and chloroquine-treated conditions, with only a very faint signal detected in the pellet (Figure R12C). Importantly, p62, GAPDH, and LC3 isoforms also partitioned primarily into the soluble fraction, indicating that no significant detergent-insoluble aggregates formed under these conditions. Combined with our earlier findings in HEK293T cells (Figure R11), these results strongly suggest that ARMS does not undergo substantial aggregation and is unlikely to be targeted by aggrephagy.

Taken together, these data demonstrate that ARMS is a physiologically relevant autophagy substrate degraded via the ATG7–GABARAP axis, and its turnover is not driven by non-specific aggregation or bridging adapters such as p62. We have now incorporated these new results into the revised manuscript as guided by the reviewer.

Figure R12. ARMS is degraded by autophagy without aggregation in primary neurons.

(A) Primary neurons were treated with ATG7 inhibitor (ATG-IN-2) for 24 h. Western blot analysis showed increased levels of endogenous ARMS and p62, along with accumulation of LC3 I.

(B) Quantification of ARMS protein levels from three independent experiments (n=3). Data are presented as mean \pm SEM, **P<0.01.

(C) Neurons treated with chloroquine (CQ) for 12 h were lysed in 1% Triton X-100 buffer. Western blot of soluble and insoluble fractions revealed that ARMS, p62, LC3-I/II, and GAPDH were almost exclusively detected in the soluble fraction, indicating the absence of aggregation.

6 • Dendritic spine assay: This is an elegant assay and a suitable one to claim physiological effects of the ARMS:GABARAP interaction. However, the key question is whether the EE mutant is disrupting the structure of ARMS ankyrin repeats and in this way destabilizes the protein (making

it perhaps proteasomal substrate)? Can the authors check the folding status of the single and double mutants in the purified N-terminal constructs of ARMS. Especially in dendritic spine formation assay, misfolded ARMS could be mistaken for the lack of GABARAP binding. What would happen if non-ARMS binding mutants of GABARAP are co-expressed with ARMS in this assay?

We thank the reviewer for raising two important points regarding the structural integrity of ARMS ankyrin-repeat mutants and the interpretation of the dendritic spine assay.

To directly assess whether the E33R and E67R mutations in ARMS compromise the folding of the ankyrin-repeat region, we purified recombinant Trx-ARMS 1-4 ARs constructs (WT, E33R, E67R, and E33R/E67R) and analyzed their secondary structure using circular dichroism (CD) spectroscopy. All variants exhibited characteristic α -helical CD profiles with highly similar spectral features compared to the WT, including minima at \sim 208 and \sim 222 nm and a positive peak near 195 nm (Figure R13), confirming that these mutations do not cause detectable structural perturbations. These data strongly support that the E33R and E67R mutations do not disrupt the fold of ARMS and that the observed loss-of-function phenotypes in the dendritic spine assay are unlikely due to misfolding. Furthermore, based on our extensive prior work on ankyrin-repeat proteins, we note that single amino acid substitutions within solvent-exposed loops or hairpin regions typically do not affect the overall fold of the repeats. We intentionally avoided introducing destabilizing core mutations when designing these ARMS variants.

Figure R13. CD spectroscopy confirms that ARMS ankyrin-repeat mutants retain α -helical secondary structure.

Purified Trx-ARMS 1-4ARs (WT), E33R, E67R, and E33R/E67R double mutant proteins were analyzed by circular dichroism. All variants exhibited characteristic α -helical spectra with double minima at \sim 208 and \sim 222 nm, and a positive peak at \sim 195 nm, indicating that the mutations do not perturb the secondary structure of the ankyrin-repeat domain.

Regarding non-ARMS binding mutants of GABARAP in dendritic spine formation assay, we previously performed additional experiments using the ARMS non-binding GABARAP mutant (L63Q) in the dendritic spine assay (please see our response to Reviewer#2 Q4 and Figure R4). In that study, we co-expressed RFP-ARMS with either wild-type GABARAP or GABARAP L63Q in primary neurons. We found that GABARAP WT significantly suppressed the ARMS-induced increase in spine density and maturation, consistent with its inhibitory role in synaptic

development. In contrast, GABARAP L63Q largely failed to exert this suppressive effect, resulting in significantly higher spine number and maturity compared to GABARAP WT (Figure R4). This phenotype closely mirrors that of the ARMS E33R/E67R mutant, supporting the conclusion that direct ARMS–GABARAP interaction is required for GABARAP to modulate ARMS function in dendritic spine regulation. These results provide strong functional evidence that the phenotypes observed in the spine assay are due to loss of specific ARMS–GABARAP interaction, rather than nonspecific effects such as misfolding or overexpression.

7• Competition between ARMS and AnkB WT peptide: It is hard to interpret the result of the ITC experiments where only one sequence of events leads to the displacement of the ARMS:GABARAP binding. Affinity of AnkB peptide is much higher, so in theory it should always displace ARMS from the GABARAP. Perhaps, it would be possible to complement this method of inhibiting ARMS:GABARAP complex formation by reconstitution of mutant forms of GABARAP in neurons. Also, would it be possible to use isolated GABARAP-binding peptides from ARMS itself in the competition assay? Would it be possible to enhance its affinity and make more like AnkB. That would be of great interest for the field, and this point should at least be mentioned in Discussion. Lastly, the GFP construct was used to bring GABARAP-competing peptides in the cells. Would it be possible to use cell-penetrating peptides instead?

We thank the reviewer for these valuable suggestions regarding the competition between ARMS and AnkB.

(1) Interpretation of ITC competition experiments

We agree that the asymmetry observed in the ITC experiments reflects the higher affinity of AnkB for GABARAP. Specifically, AnkB can displace ARMS from the ARMS–GABARAP complex, whereas ARMS fails to displace AnkB from the AnkB–GABARAP complex, yielding no detectable signal. This supports our use of AnkB as an effective dominant competitor in both *in vitro* and cellular assays.

(2) Rationale for using AnkB peptide instead of GABARAP mutants

We appreciate the reviewer's suggestion to reconstitute GABARAP mutants in neurons as an alternative means to disrupt the ARMS: GABARAP interaction. While reconstituting GABARAP mutants in neurons is a valid approach, we preferred the AnkB peptide for temporal and functional specificity. Mutant GABARAP may interfere with other pathways, whereas peptide-based inhibition offers a more targeted and reversible strategy. Our prior study validated this approach in neurons^[16] and in animals^[15]. The higher affinity of AnkB also ensures robust displacement of ARMS.

(3) ARMS-derived peptides and affinity enhancement

We appreciate the reviewer's suggestion to explore ARMS-derived GABARAP-binding peptides. However, unlike canonical linear LIR motifs, ARMS engages GABARAP through a nonlinear interface involving loop tips from two adjacent ankyrin repeats that simultaneously occupy HP1 and HP2. This interaction mode makes it difficult to isolate a short linear peptide that faithfully

recapitulates GABARAP binding. Indeed, our attempts to express minimal constructs (e.g., ARMS ANK1–2) did not yield stable proteins, limiting their use in competition assays. These structural constraints likely explain why ARMS does not conform to the typical LIR peptide paradigm and further underscore the novelty of its GABARAP recognition mechanism.

(4) Potential for enhancing ARMS–GABARAP affinity

We also appreciate the idea of engineering ARMS to achieve higher affinity for GABARAP. However, this is unlikely to be feasible. AnkB achieves its exceptionally high affinity through both a canonical LIR motif and a C-terminal α -helical extension that creates an expanded interaction surface. In contrast, ARMS relies on a more limited and structurally constrained surface formed by ankyrin-repeat loops. This fundamental difference in interface architecture likely imposes an upper limit on the achievable binding affinity for ARMS.

(5) GFP fusion versus cell-penetrating peptides

Regarding the mode of intracellular delivery, we thank the reviewer for suggesting the use of cell-penetrating peptides. Actually, we have successfully used a cell-penetrating AnkB peptide to access intracellular LC3B in our previous study^[15]. However, due to concerns with dose control, transient effects, and cytotoxicity, CPPs are less suited for long-term assays such as spine maturation. In contrast, GFP-fusion constructs provide stable, sustained expression and reliable subcellular localization, as demonstrated in our current experiments. Therefore, we selected this approach based on both efficacy and practical considerations.

Reference

- [1] Scholz-Starke J, Cesca F. Stepping Out of the Shade: Control of Neuronal Activity by the Scaffold Protein Kidins220/ARMS. *Frontiers in Cellular Neuroscience*, 2016, 10:68.
- [2] Sánchez-Ruiloba L, Cabrera-Poch N, Rodríguez-Martínez M, et al. Protein kinase D intracellular localization and activity control kinase D-interacting substrate of 220-kDa traffic through a postsynaptic density-95/discs large/zonula occludens-1-binding motif. *Journal of Biological Chemistry*, 2006, 281(27): 18888-18900.
- [3] Yi-Hua Liao, June-Tai Wu, et al. ARMS-NF- κ B signaling regulates intracellular ROS to induce autophagy-associated cell death upon oxidative stress. *iScience*, 2023, 26(2): 106005.
- [4] Richards J, Dorand M F, et al. Significantly higher rates of KIDINS220 polymorphisms in patients with obesity and end-stage renal disease. *Obesity Pillars*, 2025, 13: 100155.
- [5] Li J, Chen L A, et al. PKD1, PKD2, and their substrate Kidins220 regulate neurotensin secretion in the BON human endocrine cell line. *Journal of Biological Chemistry*, 2008, 283(5): 2614-2621.
- [6] Neubrand VE, Cesca F, Benfenati F, Schiavo G. Kidins220/ARMS as a functional mediator of multiple receptor signalling pathways. *J Cell Sci*. 2012; 125:1845–1854.
- [7] Benoit BO, Savarese T, Joly M, Engstrom CM, Pang L, Reilly J, Recht LD, Ross AH, Quesenberry PJ. Neurotrophin channeling of neural progenitor cell differentiation. *J Neurobiol*. 2001; 46:265–280.
- [8] Lipsky RH, Marini AM. Brain-derived neurotrophic factor in neuronal survival and behavior-related plasticity. *Ann NY Acad Sci*. 2007; 1122:130–143.
- [9] Higuero A M, Sanchez-Ruiloba L, Doglio L E, et al. Kidins220/ARMS modulates the activity of microtubule-regulating proteins and controls neuronal polarity and development. *Journal of Biological Chemistry*, 2010, 285(2): 1343-1357.
- [10] Gamir-Morralla A, Belbin O, Fortea J, et al. Kidins220 correlates with Tau in alzheimer's disease brain and cerebrospinal fluid. *Journal of Alzheimer's Disease*, 2016, 55(4): 1327-1333
- [11] Liao Y H, Hsu S M, Huang P H. ARMS depletion facilitates UV irradiation-induced apoptotic cell death in melanoma. *Cancer research*, 2007, 67(24): 11547-11556.
- [12] Jung H, Shin J-H, Park Y-S, Chang M-S. 2014. Ankyrin repeat-rich membrane spanning (ARMS)/KIDINS220 scaffold protein regulates neuroblastoma cell proliferation through p21. *Mol Cells* 37:881–887.
- [13] Wang C, Wei Z, Chen K, et al. Structural basis of diverse membrane target recognitions by ankyrins. *Elife*, 2014, 3: e04353.
- [14] Tseng W C, Jenkins P M, Tanaka M, et al. Giant ankyrin-G stabilizes somatodendritic GABAergic synapses through opposing endocytosis of GABAA receptors. *Proceedings of the National Academy of Sciences*, 2015, 112(4): 1214-1219.
- [15] Li J, Zhu R, Chen K, et al. Potent and specific Atg8-targeting autophagy inhibitory peptides from giant ankyrins. *Nature chemical biology*, 2018, 14(8): 778-787.
- [16] Ye J, Zou G, Zhu R, et al. Structural basis of GABARAP-mediated GABAA receptor trafficking and functions on GABAergic synaptic transmission. *Nat Commun* 12: 297. (2021)

Dear Dr. Wang,

There are still some outstanding corrections that need to be addressed before I can accept your manuscript for publication. Would you please:

- define the error bars in the legend of figure 6D,
- define the white arrows in the legend of Figure eV5 H and I,
- define the inset boxes in EV5 G and ensure the zoomed areas correspond to the indicated boxes,
- rename the materials and methods section as 'Methods', and
- provide me with a two sentence summary and bullet points that summarise your work.

Yours sincerely,

William Teale

William Teale, PhD
Editor
The EMBO Journal
w.teale@embojournal.org

All editorial and formatting issues were resolved by the authors.

Dear Dr. Wang,

I am pleased to inform you that your manuscript has been accepted for publication in the EMBO Journal.

Congratulations on a really exciting article!

Best wishes,

William Teale

William Teale, PhD
Editor
The EMBO Journal
w.teale@embojournal.org

Please note that it is The EMBO Journal policy for the transcript of the editorial process (containing referee reports and your response letters) to be published as an online supplement to each paper. If you should prefer removal of any referee-only figures included in the point-by-point response(s), e.g. because they may still be used for future publication or because they have been reproduced from published work by others, please do let us know immediately via response email.

More information is available here: https://www.embopress.org/transparent-process#Review_Process
